



# Climate pathways behind phytoplankton-induced atmospheric warming

Rémy Asselot [1], Frank Lunkeit [2], Philip Holden [3], and Inga Hense [1]

[1]Institute for Marine Ecosystem and Fishery Science, Center for Earth System Research and Sustainability, University of Hamburg, Hamburg, Germany
[2]Meteorological Institute, University of Hamburg, Hamburg, Germany
[3]Environment, Earth and Ecosystems, The Open University, Walton Hall, Milton Keynes, MK7 6AA, UK

**Correspondence:** Rémy Asselot (remy.asselot@uni-hamburg.de)

**Abstract.** We investigate in which ways marine biologically-mediated heating increases the surface atmospheric temperature. While the effects of phytoplankton light absorption on the ocean have gained attention over the past years, the impact of this biogeophysical mechanism on the atmosphere is still unclear. Phytoplankton light absorption warms the surface of the ocean with consequences for the air-sea heat exchange and $CO_2$ flux. We focus on the ocean-atmosphere interface and study

the importance of air-sea heat exchange versus air-sea $CO_2$ flux. To shed light on the role of phytoplankton light absorption on the surface atmospheric temperature, we performed different simulations with the EcoGENIE Earth system model. We configure the model without a seasonal cycle and, if not stated otherwise, the atmospheric $CO_2$ concentration is allowed to evolve freely. The climate pathways examined are: heat exchange, dissolved $CO_2$, solubility of $CO_2$, and sea-ice covered area. Overall we show that the air-sea $CO_2$ exchange has a larger effect on the biologically-induced atmospheric warming than the

air-sea heat flux. Moreover, we notice that the freely evolving solubility of $CO_2$ has a cooling effect on the surface atmospheric temperature.

## 1 Introduction

Previous studies have shown that marine biota can modify the light penetration in the ocean with consequences on the atmo-

spheric temperature and on the climate system (Shell et al., 2003; Wetzel et al., 2006; Gnanadesikan and Anderson, 2009). Using an Earth system model (ESM) of intermediate complexity, we identify and compare the climate pathways behind the changes in atmospheric temperature due to phytoplankton light absorption.

Marine biota and phytoplankton play a major role in the absorption of light and therefore in the vertical distribution of heat

in the upper layers of the ocean (Kowalczuk et al., 2019). Indeed, observational evidence support the hypothesis that chlorophyll increases the upper ocean heat uptake. For instance, satellite observations show that phytoplankton blooms can cause an





increase of sea surface temperature (SST) of 1.5°C (Kahru et al., 1993). Furthermore, previous remote sensing data indicate an increase in local SST of 4.5°C on a 4 day-timescale due to the presence of phytoplankton blooms (Capone et al., 1998). Recent high-resolution in situ observations in the Indo-West Pacific Ocean indicate large anomalies of temperature of 0.95°C

in the uppermost skin layer of the ocean when large phytoplankton blooms appear (Wurl et al., 2018). However, all these observations are either on a short time scale or in a geographically limited area. To study the larger-scale impact of phytoplankton light absorption and its varying magnitude, Earth system models are employed.

Different models with different complexity are used to study the effect of phytoplankton light absorption. For instance, us-

ing ocean-only (Anderson et al., 2007) or general circulation models (Murtugudde et al., 2002; Lengaigne et al., 2007; Löptien et al., 2009), several studies focusing on the tropical Pacific Ocean report an increase of SST between 0.5-2°C. The same magnitude of ocean warming is reported with a general circulation model focusing on the Arctic Ocean (Lengaigne et al., 2009). These changes in ocean temperature have an impact on the nutrient availability and the biogeochemical properties of the ocean (Manizza et al., 2008; Asselot et al., 2021). A warming of the surface of the ocean induced by marine biota also has conse-

quences on the overall climate system. Patara et al. (2012) find that an increase of SST due to phytoplankton light absorption increases the atmospheric humidity content thereby increasing the greenhouse effect and the atmospheric temperature by up to 0.5°C. Furthermore, phytoplankton can amplify locally the seasonal cycle of the lowest atmospheric layer temperature by 1K (Shell et al., 2003). Moreover, Shell et al. (2003) indicate that the climate effect of phytoplankton can even extend through the troposphere in mid-latitude regions, influencing the Walker and Hadley circulation.


It is therefore known that phytoplankton light absorption has a non-negligible role on the atmospheric temperature but which climate pathways are the most important behind this warming is still unclear. Phytoplankton light absorption affects the surface atmospheric temperature via two climate pathways. First, various modeling studies suggest that biologically-induced surface water heating can increase the air-sea heat exchange (Capone et al., 1998; Oschlies, 2004; Wetzel et al., 2006) with con-

sequences on the formation of tropical storms and monsoons in the Arabian Sea (Sathyendranath et al., 1991). Second, the solubility of gases and thus also the air-sea $CO_2$ exchange is affected by phytoplankton light absorption. For instance, Manizza et al. (2008) study the impact of this biogeophysical mechanism on the air-sea flux of $CO_2$ and find that phytoplankton light absorption has a small outgassing effect on a global scale with high regional fluctuations.

However, none of these studies have analyzed and compared the changes in air-sea heat and $CO_2$ exchange due to phytoplankton light absorption. To shed light on the biologically-induced atmospheric warming, we use a recent Earth system model of intermediate complexity (Claussen et al., 2002). The model is called EcoGENIE (Ward et al., 2018) where we implemented phytoplankton light absorption in an earlier study (Asselot et al., 2021). We use the same model setup to determine now the importance of this biogeophysical mechanism on biologically-induced atmospheric warming. We conduct several simulations

to determine the importance of the climate pathways behind the atmospheric warming. We consider two different biologically-induced changes: a change in air-sea heat and air-sea $CO_2$ exchange rates (Fig. 1). The air-sea $CO_2$ exchange can be influenced





by the dissolved $CO_2$ in the ocean in three different ways: through 1) the biological pump as a result of phytoplankton light absorption that affects the marine biogeochemical cycles (Manizza et al., 2008; Asselot et al., 2021), 2) the solubility of $CO_2$ due fluctuations of SST and 3) sea-ice formation and resulting sea-ice extent altering the air-sea $CO_2$ flux.


The paper is organized as follow: In section 2, we describe the components of the model, the light absorption scheme and the air-sea exchanges. In section 3, we describe the simulations and the modeling strategy. In section 4, we report several sensitivity analyses of the climate system with EcoGENIE. In section 5, we present our results and detail the changes in both

oceanic and atmospheric properties. In section 6, we conclude by commenting on the role of this biogeophysical mechanism in the atmospheric warming.

## 2 Model description

Our motivation is to study the interactions between the marine ecosystem, the biogeochemistry, the biogeophysics and the climate system. These interactions are computationally expensive in high-resolution models therefore we used an Earth system

model of intermediate complexity (Claussen et al., 2002). The Earth system model employed is the Grid-ENabled Integrated Earth system model (GENIE) (Lenton et al., 2007) composes of several modules describing the dynamics of the Earth system (Fig. 2). This model has been previously calibrated and compared to observations several times (Edwards and Marsh, 2005; Lenton et al., 2006; Ridgwell et al., 2007; Marsh et al., 2011). GENIE is widely used to study past climate and changes in the carbon cycle over geological times (Greene et al., 2019; Adloff et al., 2020). Furthermore, GENIE has been used to

demonstrate that the sensitivity of atmospheric $CO_2$ is mainly explained by the organic carbon pump (Cameron et al., 2005). We use the carbon-centric version (cGENIE) that has been previously employed to study past mass extinction (Alvarez et al., 2019), the climate system (Ödalen et al., 2018) or biogeochemistry processes (Meyer et al., 2016). GENIE is associated with the new ecosystem component (ECOGEM) to form the recent EcoGENIE model (Ward et al., 2018). EcoGENIE has been used to determine the link between the marine plankton ecosystem and various past climate scenarios (Wilson et al., 2018)

with focus on phosphorus inventory (Reinhard et al., 2020). For our study, the model combines different components including ocean hydrodynamics, atmosphere, sea-ice, ocean biogeochemistry and marine ecosystem component. The efficient numerical terrestrial scheme (Williamson et al., 2006) is not used in this study, so the land surface is essentially passive. We use the same configuration as described in detail by Asselot et al. (2021) and thus only briefly explain the individual model components.

### 2.1 Modules

#### 2.1.1 The physical components

The physics of the model contains a frictional-geostrophic ocean circulation (GOLDSTEIN), coupled to a 2D energy-moisture balance model of the atmosphere (EMBM) and a thermodynamic sea-ice model (GOLDSTEINSEAICE) (Edwards and Marsh,





2005; Marsh et al., 2011). Heat and moisture are exchanged between the three components and act as a coupling strategy.

The oceanic component calculates the horizontal and vertical redistribution of heat, salinity and biogeochemical elements via
advection, convection and mixing. The ocean module is configured on a $36 \times 36$ horizontal grid. The horizontal grid is uniform
in longitude and uniform in sine latitude, giving $\sim3.2°$ latitudinal increments at the equator increasing to $19.2°$ in the highest
latitude. This horizontal grid has been used for previous biogeochemical simulations (Cameron et al., 2005; Colbourn, 2011).
We consider 32 vertical oceanic layers increasing logarithmically from 29.38 m for the surface layer to 456.56 m for the deepest
layer. This vertical resolution has already been used to study the relative importance of biogeophysical and biogeochemical
mechanisms on the climate system (Asselot et al., 2021).

The atmospheric component is based closely on the UVic Earth system model (Weaver et al., 2001). The prognostic variables
are atmospheric temperature and specific humidity. Precipitation removes instantaneously all moisture corresponding to an
excess above a relative humidity threshold.

The sea-ice component solves the equation for part of the ocean covered by sea-ice. The prognostic variables are ice thickness
and ice areal fraction. The transport of sea-ice includes sources and sinks of these variables. The growth or decay of sea ice
depends on the net heat flux into the ice. The dynamics in this module consist of advection by currents and diffusion.

### 2.1.2  Ocean biogeochemistry component

The biogeochemical module (BIOGEM) represents the transformation and spatial redistribution of biogeochemical tracers
(Ridgwell et al., 2007). The state variables are inorganic resources and organic matter. The biological uptake is represented
by an implicit biological community: nutrients are directly converted into organic matter via an uptake rate. The biological
uptake is limited by light, temperature and nutrient availability. Organic matter is partitioned into dissolved and particulate
phases (DOM and POM). The model includes iron (Fe) and phosphate ($PO_4$) as limiting nutrients. Similar to Asselot et al.
(2021), we do not consider nitrate ($NO_3^-$) here. Furthermore, BIOGEM calculates the air-sea $CO_2$ and $O_2$ exchange. The value
of atmospheric $CO_2$ predicted by BIOGEM is used as input for the radiative scheme of the atmospheric component, thus
providing climate feedback.

### 2.1.3  Ecosystem component

The marine ecosystem component (ECOGEM) represents the marine plankton community and associated interactions in the
ecosystem (Ward et al., 2018). The biological uptake in ECOGEM replaces the BIOGEM uptake calculation and is limited
by light, temperature and nutrient availability. Plankton biomass and organic matter are subject to processes such as resource
competition and grazing before being passed to DOM and POM. The ecosystem is divided into different plankton functional
types (PFTs) with specific traits. Furthermore, each PFT is sub-divided into size classes with specific size-dependent traits.
We consider two classes of PFTs: phytoplankton and zooplankton. Phytoplankton is characterized by nutrient uptake and
photosynthesis whereas zooplankton is characterized by predation traits. Zooplankton grazing depends on the concentration
of prey biomass availability, with predominantly grazing on preys that are 10 times smaller than themselves. Each population
is associated with biomass state variables for carbon, phosphate and chlorophyll. The production of dead organic matter is





a function of mortality and messy feeding, with partitioning between non-sinking dissolved and sinking particulate organic matter. Finally, plankton mortality is reduced at very low biomass such that plankton cannot become extinct.

## 2.2 Light absorption in the ocean

The implementation of phytoplankton light absorption in EcoGENIE is the same as the scheme described in Asselot et al. (2021) and is a coupling between Eq. 1 and Eq. 2. For a simplification issue, in our model configuration, the incoming shortwave radiation does not vary seasonally. We look at long-term changes in the climate system therefore the absence of a seasonal cycle does not affect our results and main findings. The presence of organic, inorganic particles and dissolved molecules restrains the light penetration in the ocean (Ward et al., 2018). The vertical light attenuation scheme is given by Eq.1:

$$I(z) = I_0 \cdot \exp(-k_w - k_{Chl} \cdot Chl_{tot}) \cdot z \tag{1}$$

where $I(z)$ is the irradiation of the full solar spectrum at depth $z$, $I_0$ is the irradiation at the surface of the ocean, $k_w$ is light absorption by clear water and inorganic particles ($0.04 \, \mathrm{m}^{-1}$), $k_{Chl}$ is the light absorption by chlorophyll ($0.03 \, \mathrm{m}^{-1} (\mathrm{mg \, Chl})^{-1}$) and $Chl_{tot}$ is the total chlorophyll concentration. The values for $k_w$ and $k_{Chl}$ are taken from Ward et al. (2018). The parameter $I_0$ is negative in the model because it is a downward flux from the sun to the surface of the ocean. We allow primary production and light to penetrate until the sixth layer of the model (221.84 m deep), which is the lower limit of the euphotic zone (Tett, 1990). In our model setup, maximum absorption occurs in the upper oceanic layer and minimum absorption occurs in the sixth layer.

Phytoplankton changes the optical properties of the ocean (Sonntag and Hense, 2011) through phytoplankton light absorption. Therefore it can cause a radiative heating and change the oceanic temperature. We implemented phytoplankton light absorption into the model following Hense (2007) and Patara et al. (2012) Eq.2:

$$\frac{\partial T}{\partial t} = \frac{1}{\rho \cdot c_p} \frac{\partial I}{\partial z} \tag{2}$$

$\partial T / \partial t$ denotes the temperature changes, $c_p$ is the specific heat capacity of water, $\rho$ is the ocean density, $I$ is the solar radiation incident at depth $z$. Part of the light absorbed is used by phytoplankton for photosynthesis and part is released in form of fluorescence and heat. However, the fluorescence form can be ignored, therefore it is assumed that the whole light absorption leads to heating of the water (Lewis et al., 1983).



## 2.3 Air-sea heat exchange

Heat is exchanged between the atmosphere, the ocean and the sea-ice components and acts as a coupling between these three modules. We detail here only the relevant fluxes for our study, the heat flux into the atmosphere. The vertically integrated atmospheric heat equation is given by Weaver et al. (2001) and Marsh et al. (2011) Eq. 3:

$$Q_{ta} = Q_{SW} \cdot C_A + Q_{LH} + Q_{LW} + Q_{SH} - Q_{PLW} \tag{3}$$

$Q_{ta}$ corresponds to the total heat flux into the atmosphere, $Q_{SW}$ is the net shortwave radiation corresponding to the solar irradiance receives from the sun and reflected by the planet's albedo, $C_A$ is a heat absorption coefficient (0.3 over the ocean, Marsh et al. (2011)), $Q_{LH}$ is the latent heat flux corresponding to phase change of a certain thermodynamic system, $Q_{SH}$ is the sensible heat flux corresponding to temperature change of a thermodynamic system, $Q_{LW}$ is the net (upward minus downward) re-emitted longwave radiation corresponding to infrared energy coming from the planet and $Q_{PLW}$ is the outgoing planetary longwave radiation.

The atmosphere loses heat through net longwave radiation, dominated by the outgoing longwave radiation, thus the total longwave heat flux ($Q_{LW} + Q_{PLW}$) is negative in the model. Furthermore, evaporative cooling of the ocean leads to a latent heat release in the atmosphere upon condensation and precipitation. Evaporated water vapour may be transported away from an oceanic source, to condense and precipitate elsewhere.

## 2.4 Air-sea CO$_2$ exchange

The atmospheric temperature depends on the atmospheric $CO_2$ concentration which is affected by the transfer of $CO_2$ between the ocean and the atmosphere. The flux of $CO_2$ across the atmosphere-ocean interface is given by Ridgwell et al. (2007) Eq. 4:

$$F_{CO_2} = k \cdot \rho \cdot (C_w - \alpha \cdot C_a) \cdot (1 - A) \tag{4}$$

$F_{CO_2}$ is the air-sea $CO_2$ flux, $k$ corresponds to the gas transfer velocity, $\rho$ is the ocean density, $C_w$ is the concentration of dissolved gas in the surface ocean, $\alpha$ is the solubility coefficient calculated from Wanninkhof (1992) and depends on the sea surface temperature and salinity, $C_a$ is the concentration of gas in the atmosphere and $A$ is the fraction of the ocean covered by sea-ice.

Phytoplankton light absorption affects the flux of $CO_2$ via the parameters $C_w$, $\alpha$ and $A$. To study precisely the flux we either prescribe these parameters in the air-sea $CO_2$ exchange calculation or let them evolve freely. To prescribe these parameters we take the values from the reference run (see below).



# 3 Model setup and simulations

During this study, we are mainly interested in the relative differences between our selected simulations. We try to simulate a realistic mean climate system but the absolute value of the climate quantities are less relevant due to the limitations of such an intermediate complexity model.

For a realistic nutrient distribution in the ocean, we performed a BIOGEM spin-up for 10,000 years. During the spin-up the atmospheric $CO_2$ concentration is fixed to 278 ppm. The simulations restart for 1,000 years after the spin-up with ECOGEM, meaning that all simulations consider marine biota. The model setup, ecosystem community and grid resolution employed are the same as in Asselot et al. (2021) (Appendix A1) except that we run the model without any seasonal cycle. The seasonal cycle is removed for technical issues, we cannot prescribe the seasonal cycle of SST but only the annually-averaged SST. The

absence of the seasonal cycle is not an issue for this study because we look at the importance of each climate pathway rather than focusing on the quantitative changes of the climate system.

The carbon cycle is closed in our simulations, meaning that there is no input of carbon through volcanic or anthropogenic activities. Only the size of the carbon reservoirs can vary. If not stated otherwise, the concentration of atmospheric $CO_2$ evolves freely in the simulations. Furthermore, all simulations are forced with the same constant flux of dissolved iron into the ocean

surface (Mahowald et al., 2006).

To study the effect of phytoplankton light absorption on the atmospheric temperature we perform seven different simulations (Fig. 3; Table 1):

– The first one, called *Bio* is the reference run and is the only simulation that does not include phytoplankton light absorption ($k_{Chl} = 0$ in Eq. 1). In this simulation, all the climate pathways evolve freely.

  – The second one, called *BioLA* is the same as the reference run but with phytoplankton light absorption implemented. In this simulation, all the climate pathways evolve freely.

  – The third simulation *HEAT* is the same as the second one except that we prescribe the atmospheric $CO_2$ concentration
only for the atmospheric temperature calculation. For a comparison with the reference run, the prescribed atmospheric $CO_2$ concentration from *Bio* is used (169 ppm). The effect of $CO_2$ on atmospheric temperature is fixed but the air-sea heat fluxes evolve freely. This simulation determines the effect of air-sea heat flux on the energy budget.

  – The fourth simulation is named *CARB* where we run the model with an uncoupled ocean-atmosphere setup. The atmospheric component is forced with the heat fluxes from the reference run and the atmospheric $CO_2$ concentration
is prescribed with the value of *BioLA*. This simulation determines the effect of phytoplankton-induced changes of atmospheric $CO_2$ concentration on the atmospheric temperature. Please note that *CARB* is well suited for studying the atmosphere properties but not to examine ocean dynamics.





- The fifth simulation is named *HCorg* and we only allow the biological pump to affect the dissolved $CO_2$. The solubility of $CO_2$ ($\alpha$ in Eq. 4) and sea-ice extent ($A$ in Eq. 4) parameters are prescribed using the respective values from *Bio*.

The $CO_2$ solubility is fixed by prescribing the SST only for this calculation. In *HCorg* air-sea heat exchange and the biological pump parameter ($C_w$ in Eq. 4) evolve freely.

- The sixth simulation is called *HCorgSI* where the biological pump and sea-ice extent affect dissolved $CO_2$. The solubility of the $CO_2$ parameter ($\alpha$ in Eq. 4) is prescribed using the value of *Bio*. In *HCorgSI* the air-sea heat exchange, the biological pump ($C_w$ in Eq. 4) and sea-ice extent ($A$ in Eq. 4) parameters evolve freely.

- The seventh and last simulation is called *HCorgSol* where the biological pump and the solubility pump affect dissolved $CO_2$ in the ocean. The sea-ice extent parameter ($A$ in Eq. 4) is prescribed using the value of *Bio*. In *HCorgSol* the air-sea heat exchange, the biological pump ($C_w$ in Eq. 4) and the $CO_2$ solubility ($\alpha$ in Eq. 4) parameters evolve freely.

## 4   Sensitivity analysis

### 4.1   Climate variability

To analyze the climate variability of the model, we perform two sensitivity analyses (Table 2). Both simulations have the same model setup, restart from the spin-up described previously but their atmospheric $CO_2$ concentration differ. The first simulation (Sensi280) has an atmospheric $CO_2$ concentration of 280 ppm while the second one (Sensi320) has an atmospheric $CO_2$ concentration of 320 ppm. Furthermore, the simulations Sensi280 and Sensi320 consider phytoplankton light absorption.

An increase of 40 ppm in atmospheric $CO_2$ concentration slightly reduces the chlorophyll concentration but these changes are negligible. The oceanic and atmospheric heat budgets are affected by the changes in atmospheric $CO_2$ concentration. Increasing the greenhouse gas concentrations increases in turn the SAT and therefore the SST due to the exchange of heat between the ocean and the atmosphere.

### 4.2   Air-sea fluxes interactions

To estimate the unique effect of each climate pathway we ensure that the heat and $CO_2$ interaction is negligible. Due to the model setup, the flux of $CO_2$ across the air-sea interface ($F_{CO_2}$; Eq. 4) depends on the SST via the Schmidt number (Wanninkhof, 1992; Ridgwell et al., 2007). We conduct two comparable sensitivity analysis and analyze the changes in $F_{CO_2}$. First, we artificially increase the SST by 1°C and do not exceed the maximum difference of SST between our simulation results (Table 3). This increase in SST only enhances $F_{CO_2}$ by $4.26 \cdot 10^{-5}$ mol/m²/yr, representing a raise of 2.58% of the total air-sea

$CO_2$ exchange. Even large SST fluctuations negligibly affect the flux of $CO_2$ at the air-sea interface. Second, the mean wind speed affects the $F_{CO_2}$ via the gas transfer velocity ($k$; Eq. 4). We increase the wind speed by 0.2 m/s, which is a comparable forcing of the artificial increase of 1°C of SST (Knutson and Tuleya, 2004). This increase in mean wind speed enhances the $F_{CO_2}$ by $1.44 \cdot 10^{-4}$ mol/m²/yr, representing an increase of 8.69% of the total air-sea $CO_2$ flux. Clearly, the changes in wind





speed are much larger than the changes in SST hence we consider that the effect of SST on the air-sea $CO_2$ exchange is small

enough to be neglected.

## 5  Global response of the climate system

In this section we present the results of the simulations on a global scale, we do not consider local patterns because we removed any seasonal cycle in our model setup. As already mentioned, the absence of the seasonal cycle is not an issue for our study because we focus on the importance of each climate pathway rather than analyzing the quantitative assessments of the climate

pathways. First, we focus on the chlorophyll biomass and sea surface temperature because phytoplankton light absorption has a direct effect on these climate variables (Oschlies, 2004; Lengaigne et al., 2007; Paulsen et al., 2018). Second, these changes in oceanic properties affect the carbon cycle (Manizza et al., 2008; Asselot et al., 2021), therefore we study the changes in atmospheric $CO_2$ concentration between the simulations. Third, phytoplankton light absorption alters the atmospheric properties (Patara et al., 2012), thus we analyze the changes in radiative heat fluxes, humidity and evaporation between the simulations.

Finally, due to changes in oceanic and atmospheric properties, the response of the surface atmospheric temperature is studied.

### 5.1  Chlorophyll biomass and sea surface temperature

Our results indicate differences of sea surface temperature (SST) and chlorophyll biomass, depending on the climate pathways included in our model setup (Table 4). The reference run *Bio* has the lowest chlorophyll biomass and a low SST while the simulation *BioLA* has the highest chlorophyll biomass and SST. As previously demonstrated, phytoplankton light absorption

increases the chlorophyll biomass and therefore the SST via shallower downward flux of organic matter and higher surface nutrient concentrations (Manizza et al., 2008; Asselot et al., 2021). The chlorophyll biomass difference between *BioLA* and *Bio* is 0.012 mgChl/m$^3$ which is in agreement with previous estimates (Manizza et al., 2005; Asselot et al., 2021). However, the global difference of SST between *BioLA* and *Bio* of only 0.08°C is lower than previous estimates (Lengaigne et al., 2009; Löptien et al., 2009; Asselot et al., 2021). This underestimation of the biologically-induced SST heating is due to non-seasonal

radiative forcing of the model. The non-seasonal radiative forcing decreases the global heat budget (Appendix B1), explaining the lower response of the SST in our study.

The chlorophyll biomass is higher while the SST is lower in *HEAT* compared to the reference simulation (Table 4). This is rather counter-intuitive and is due to changes in oceanic circulation between these two simulations. For instance, the maxi-

mum Atlantic overturning circulation is 8.6 Sv in *HEAT* while it is 7.6 Sv in *Bio*. The stronger overturning circulation in *HEAT* increases the concentration of surface nutrients. Specifically the surface $PO_4$ concentration is about 0.21 $\mu$mol/kg in *HEAT* while it is about 0.19 $\mu$mol/kg in *Bio*. The higher surface $PO_4$ concentration in *HEAT* explains the higher chlorophyll biomass in this simulation compared to the reference simulation. The changes in the strength of the circulation explain as well the lower SST in *HEAT* compared *Bio*. The stronger oceanic circulation in *HEAT* leads to a more important redistribution of heat along

the water column, explaining the surface cooling and the warming of the bottom water. Our results indicate that the bottom





water temperature in *HEAT* is 3.57°C while it is 3.09°C in *Bio*.

The simulation *HCorg*, *HCorgSI* and *HCorgSol* have a higher chlorophyll biomass and SST than the reference run. Fur-thermore, the chlorophyll biomass and the SST are similar between the simulation *HCorg* and *HCorgSI* indicating that the

changes in sea-ice extent due to phytoplankton light absorption do not affect these climate variables (Appendix C1). In addi-tion, the chlorophyll biomass and SST are higher in *HCorg* than in *HCorgSol*, indicating that the solubility factor has a negative feedback on these climate variables. Between these two simulations, the only difference is the $CO_2$-solubility factor that can evolve freely in *HCorgSol*. In the simulation *HCorg*, the SST for the calculation of the solubility of $CO_2$ is prescribed using the values of the reference run. The SST in the reference run is lower than the SST in *HCorgSol*. Considering the physical

and chemical properties of the ocean, a low SST increases the solubility of $CO_2$ (Wanninkhof, 1992). Therefore, the $CO_2$ solubility is reduced in *HCorgSol* compared to *HCorg*, due to the higher SST in *HCorgSol*. For instance, our results indicate that on a global scale, the oceanic $CO_2$ concentration is 27.200 $\mu$mol/kg in *HCorgSol* while it is 27.213 $\mu$mol/kg in *HCorg*. These changes in carbon cycle between the simulations affect the others biogeochemical cycles via the nutrient ratios (Ward et al., 2018). As a consequence, the surface $PO_4$ concentration is about 0.216 $\mu$mol/kg in *HCorg* and about 0.214 $\mu$mol/kg in

*HCorgSol*. The higher $PO_4$ concentration at the surface in *HCorg* leads to the higher chlorophyll biomass and higher SST due to phytoplankton light absorption compared to *HCorgSol*.

## 5.2 Atmospheric properties

The oceanic properties differ between the simulations, thus we expect differences between the atmospheric properties in each simulation. First, we compare the atmospheric $CO_2$ concentration, then the heat fluxes, the evaporation, the specific humidity

and finally the surface atmospheric temperature.

### 5.2.1 Atmospheric $CO_2$ concentration

In all the simulations considering phytoplankton light absorption, the atmospheric $CO_2$ concentration is higher than in the ref-erence run (Table 5). The atmospheric $CO_2$ concentration is the lowest in *Bio* while it is the highest in *BioLA*, with a difference of 9 ppm. The difference of $CO_2$ concentration between the simulations *BioLA* and *Bio* is lower than previous estimate (Asselot

et al., 2021) and is due to the non-seasonal cycle forcing (Appendix B1). As already described in Asselot et al. (2021), the higher atmospheric $CO_2$ concentration in *BioLA* is mainly explained by lower $CO_2$ solubility due to a higher SST.

In *HEAT*, the atmospheric $CO_2$ concentration is prescribed only in the atmospheric temperature calculation, therefore the atmospheric $CO_2$ concentration can vary due to changes in dissolved oceanic $CO_2$, solubility and sea-ice extent, and therefore

affect the other climate variables. The atmospheric $CO_2$ concentration in *HEAT* is slightly higher than in *Bio*. The chlorophyll biomass is more important in *HEAT* than in the reference simulation, indicating a higher amount of organic matter and there-fore a more important remineralization rate in the ocean. During the remineralization process, $CO_2$ is produced (Sarmiento and Gruber, 2006), therefore the higher remineralization rate in *HEAT* increases the dissolved $CO_2$ concentration. On a global





scale, our results indicate that the surface dissolved $CO_2$ is about 6.354 mol/kg in *HEAT* while it is 6.302 mol/kg in *BIO*.
The more important dissolved $CO_2$ concentration in *HEAT* increases the air-sea $CO_2$ flux and therefore the atmospheric $CO_2$
concentration (see Eq. 4).

The atmospheric $CO_2$ concentration in *CARB* is similar to the one in *BioLA* because we prescribed the value against the
one in *BioLA*.


The simulations *HCorg*, *HCorgSI* and *HCorgSol* have a higher atmospheric $CO_2$ concentration than the reference run. This is
again not surprising because these simulations consider phytoplankton light absorption which increase the atmospheric $CO_2$
concentration (Asselot et al., 2021). The atmospheric $CO_2$ concentration between *HCorg* and *HCorgSI* is similar due to the
similar sea-ice extent (Appendix C1) and sea-ice thickness, the sea-ice does not have an impact on the atmospheric $CO_2$ con-
centration. The slightly higher atmospheric $CO_2$ concentration in *HCorgSol* compared to *HCorg* is due to changes in $CO_2$
solubility between these two simulations. As described above, the $CO_2$ solubility is lower in *HCorgSol* compared to *HCorg*.
As a consequence, the air-sea $CO_2$ flux is higher in *HCorgSol* compared to *HCorg*, leading to a slightly higher atmospheric
$CO_2$ concentration in *HCorgSol* (Eq. 4).

### 5.2.2 Heat fluxes

The air-sea heat flux is divided into the net shortwave radiation, the net re-emitted longwave radiation, the sensible heat flux
and the latent heat flux (Fig. 4). The simulations *HCorg* and *HCorgSI* have exactly the same heat fluxes because these simu-
lations are identical in all points (Appendix C1). Furthermore, the simulations *BioLA* and *HCorgSol* also have the same heat
fluxes. The only difference between these two simulations is the prescribed and different sea-ice extent for the calculation of
the air-sea $CO_2$ flux. This change in air-sea $CO_2$ flux does not alter the air-sea heat flux explaining the identical radiative heat
fluxes between *BioLA* and *HCorgSol*. Finally, the heat fluxes between *CARB* and *Bio* are identical because we prescribed the
heat fluxes in *CARB* with the values of *Bio*.

The net shortwave heat flux is divided in two: the incoming shortwave radiation from the sun entering the atmosphere and
the outgoing reflected shortwave radiation leaving the atmosphere. Figure 4a shows that the net shortwave heat flux is identical
for all the simulations and is positive. The positive values indicate that net shortwave heat flux is dominated by the flux entering
the system, the incoming radiation. The incoming shortwave radiation from the sun is always identical between simulations,
therefore identical net shortwave heat flux implies that the outgoing reflected shortwave radiation is as well the same between
simulations due to the treatment of shortwave radiation in the atmosphere given by Weaver et al. (2001).

The net longwave heat flux is negative for all simulations pointing out that this flux is dominated by the upward longwave
radiation leaving the atmosphere (Fig. 4b). A higher negative value of net longwave heat flux indicates a higher loss of heat
in outer space. The simulations *Bio* and *CARB* have the highest net longwave heat flux while the simulation *HEAT* has the





lowest heat flux, indicating that the simulation *HEAT* loses more heat than the others simulations. The higher heat loss in the simulation *HEAT* is due to a reduced amount of greenhouse gases, precisely a low specific humidity (Table 6) and atmospheric

$CO_2$ concentration (Table 5). The lower amount of greenhouse gases in the atmosphere permits a higher loss of heat outside the atmosphere. All the simulations considering phytoplankton light absorption, except *CARB* where the heat fluxes are prescribed, have a higher net longwave heat flux compared to *Bio*, which is rather predictable because this biogeophysical mechanism is an additional heat source.

The sensible heat flux depends on the atmospheric and oceanic temperature (Fanning and Weaver, 1996; Weaver et al., 2001). The sensible heat flux increases when the atmospheric temperature decreases and when the oceanic temperature increases. For the simulation *HEAT*, the sensible heat flux is the highest (Fig. 4c) because the atmospheric temperature is the lowest (Table 7). In contrast, the sensible heat flux is the lowest for the simulation *BioLA* because the gradient of temperature between the ocean and the atmosphere is low. The sensible heat flux in *HCorg* and *HCorgSI* are close to the sensible heat flux of the reference run

because their air-sea temperature gradients are almost similar.

The global mean latent heat flux (Fig. 4d) depends mainly on the global mean precipitation rate (Weaver et al., 2001). The precipitation rate between *BioLA*, *HCorg*, *HCorgSI* and *HCorgSol* are almost similar (Appendix D1) explaining the similar latent heat fluxes between these simulations. The precipitation rate in *HEAT* is higher than in *Bio*, explaining the higher latent

heat flux in *HEAT*. Furthermore, the reference run and *CARB* have the smallest latent heat flux due to the small precipitation rate for these simulations.

### 5.2.3   Specific humidity and evaporation

The specific humidity and the evaporation in *BioLA* and *HCorgSol* are similar and the same is true between the simulations *HCorg* and *HCorgSI* (Table 6). Furthermore, the specific humidity and evaporation are the lowest in the reference simulation

due to the lowest latent heat flux in this simulation. Including phytoplankton light absorption changes the heat budget, specifically increasing the latent heat flux and therefore increasing the specific humidity and evaporation, which is consistent with Oschlies (2004); Lengaigne et al. (2009). In *BioLA* the specific humidity increases by 0.5% and the evaporation increases by 0.11% compared to the reference run, which is lower than previous values (Patara et al., 2012). The different estimates between our results and Patara et al. (2012) may come from the non-seasonal cycle in our model setup, changing the heat budget

and therefore the specific humidity and evaporation rate. Moreover, the specific humidity in *HEAT* is lower than in *BioLA* due to the lower latent heat flux in the simulation *HEAT*. The evaporation depends on several pathways and one of the most important is the humidity in the atmosphere (Peixoto and Oort, 1992), the lower is the humidity the higher is the evaporation rate. As a consequence, the evaporation is higher in *HEAT* than in the simulation *BioLA*. Furthermore, the specific humidity and the evaporation increase when the atmospheric temperature rises as well (Peixoto and Oort, 1992). The specific humidity

and evaporation is higher in the simulations *CARB* compared to *BioLA* because the surface atmospheric temperature is higher *CARB* (Table 7). The specific humidity and evaporation in *HCorg* and *HCorgSI* are slightly lower than in *BioLA* because the



latent heat flux in *HCorg* and *HCorgSI* is slightly lower. Once the $CO_2$ solubility factor is considered (simulation *HCorgSol*), the values of the specific humidity and evaporation are similar to the values in *BioLA*. This is rather not surprising because the heat fluxes between *HCorgSol* and *BioLA* are identical.

### 370  5.2.4   Surface atmospheric temperature

The difference of atmospheric properties between simulations lead indubitably to changes of the surface atmospheric temperature (Fig. 5; Table 7). First of all, the reference simulation *Bio* has the lowest SAT because it doesn't include the additional heat source coming from the phytoplankton light absorption mechanism. The global difference of SAT between *BioLA* and *Bio* is 0.14°C which is lower than previous estimates (Shell et al., 2003; Patara et al., 2012; Asselot et al., 2021). The small difference of SAT compared to previous studies is clearly due to our model setup, with a non-seasonal solar radiation forcing.

The lower SAT in *HEAT* compared to *Bio* is due to several reasons. Even if *HEAT* considers phytoplankton light absorption, we show that the SST in *HEAT* is lower than in the reference run. Furthermore, for the SAT computation, the atmospheric $CO_2$ concentration is identical between *Bio* and *HEAT* and the specific humidity is slightly higher in *HEAT*. Therefore the greenhouse gas effect between these two simulations is rather similar. However, the global net longwave heat flux decreases by $\sim$0.2 W/m$^2$ in *HEAT*, leading to a cooling of the atmosphere. The combination of these different reasons explains the slightly lower SAT in *HEAT* compared to the reference simulation.

For the simulation *CARB*, the concentration of greenhouse gases (atmospheric $CO_2$ and specific humidity) is higher than in *Bio* while the air-sea heat fluxes are identical. As a consequence, more heat is trapped in the atmosphere and the SAT increases by 0.71°C compared to the reference run.

The sea-ice extent and thickness are identical between *HCorg* and *HCorgSI* (Appendix C1), resulting in identical response of the climate system and identical SAT. The specific humidity and the atmospheric $CO_2$ concentration are slightly higher in *HCorg* compared to *Bio*. This slightly higher greenhouse gas concentration leads to a small increase in SAT of *HCorg* compared to *Bio*.

In *HCorgSol* the atmospheric $CO_2$ concentration and the specific humidity are higher than in the reference simulation. However, the sensible heat flux and the net longwave heat flux are lower in *HCorgSol*. Even if the greenhouse gases concentrations are higher, the reduced in air-sea heat fluxes lead to a slight decrease in SAT in the simulation *HCorgSol* compared to *Bio*.

## 6   Conclusions

To study how phytoplankton light absorption alters the surface atmospheric temperature via air-sea heat and $CO_2$ exchange, we use the EcoGENIE model (Ward et al., 2018). For the first time, we compare the role of these individual fluxes and





quantify their influence on the biologically-induced atmospheric warming. We show that without any seasonality and with all
the climate pathways included, the surface atmospheric temperature increases by 0.14°C due to phytoplankton light absorption.
As suggested by Capone et al. (1998); Oschlies (2004); Wetzel et al. (2006), phytoplankton light absorption changes the air-sea
heat flux. Our results indicate that when only this air-sea interaction is considered, the atmosphere cools by 0.02°C compared
to a simulation without the biogeophysical mechanism. Moreover, when only the air-sea $CO_2$ exchange is considered, the
atmospheric temperature increases by 0.71°C. Clearly, our results indicate that the air-sea $CO_2$ exchange has a more important
effect than the air-sea heat flux on the phytoplankton-induced warming of the atmosphere. With our model setup, the sea-ice
extent and thickness slightly vary between simulations, therefore sea-ice processes hardly affect the air-sea $CO_2$ flux and thus
the climate system. Moreover, including the solubility pathway changes the heat fluxes, specifically reducing the sensible heat
flux and the net longwave heat flux compared to the reference simulation. As a consequence, this climate pathway has a negative
effect on the atmospheric temperature. To conclude, phytoplankton light absorption influences the climate pathways at the
ocean-atmosphere interface, particularly the air-sea $CO_2$ exchange that is important for the phytoplankton-induced atmospheric
warming. For future work, more studies with higher complexity models are necessary to make quantitative assessments rather
than qualitative assessments as in our study. For instance, a model with a dynamic atmosphere such as PLASIM-GENIE
(Holden et al., 2016) could be a good aspiration to complete our study. Observations and modeling studies indicate that
positively buoyant phytoplankton groups, such as cyanobacteria, are important to study the climate system (Sonntag and Hense,
2011; Paulsen et al., 2018; Wurl et al., 2018). Implementing these microorganisms to assess our research question could be a
beneficial follow-up of our study. Moreover, similar simulations must be conducted with a seasonal variation of the shortwave
radiation to better understand the role of phytoplankton in the climate system.

*Code availability.*  The code for the model is hosted on GitHub and can be obtained by cloning or downloading: https://zenodo.org/record/4733736.
The configuration file is named "RA.ECO.ra32lv.FeTDTL.36x36x32" and can be found in the directory "EcoGENIE_LA/genie-main/configs".
The user-configuration files to run the experiments can be found in the directory "EcoGENIE_LA/genie-userconfigs/RA/Asselotetal_BG".
Details of the code installation and basic model configuration can be found on a PDF file (https://www.seao2.info/cgenie/docs/muffin.pdf).
Finally, section 9 of the manual provides tutorials on the ECOGEM ecosystem model.



**Appendix A: Plankton functional types**

We base our ecosystem community on the community described by Ward et al. (2018). However, instead of using 16 plankton functional types (PFTs) we only use 2 PFTs: one phytoplankton group and one zooplankton group (Appendix A1). We show
that the complexity of the ecosystem does not have an important impact on the climate system compared to the effect of phytoplankton light absorption (Asselot et al., 2021). Therefore we reduced the ecosystem complexity to increase the computational time of the model.

**Appendix B: Seasonal and non-seasonal cycle**

We compare two model simulations with phytoplankton light absorption. The model setups are similar except that we switched
off the seasonal cycle in one simulation. Turning off the seasonal cycle decreases the SST by 0.77°C. Furthermore, the difference of atmospheric $CO_2$ concentration is 6 ppm. This difference is due to different SST and therefore $CO_2$ solubility between these simulations. These results indicate that switching off the seasonal cycle damps the response of the climate system to phytoplankton light absorption. Our results without seasonality indicate that the difference of SST between BioLA and Bio is 0.14°C. Similar simulations have been conducted with a seasonal cycle and the SST difference is 0.33°C (Asselot et al., 2021).
The absence of a seasonal cycle reduces the difference of SST between the simulations with and without phytoplankton light absorption.

**Appendix C: Sea-ice**

The global sea-ice cover and the global sea-ice area between the simulations HCorg and HCorgSI are identical, explaining their identical climate state. Moreover, the variation of sea-ice between all simulations is small. The maximum global sea-ice
cover of 1.42% occurs between the simulations CARB and HCorgSol.

**Appendix D: Precipitation**

Slight fluctuations in precipitation are visible in the Appendix D1. First of all, the precipitation between *BioLA* and *HCorgSol* are similar and the same is true for the precipitation between *HCorg* and *HCorgSI*. The precipitation rate is the highest in the simulation *BioLA* due to the important specific humidity. In contrast, *HEAT* has a low specific humidity explaining the lowest
precipitation rate for this simulation.





*Author contributions.* All authors designed and developed the concept of the study. RA performed the analysis of the model outputs with inputs from IH. RA drafted the initial version of the manuscript in collaboration with IH. All co-authors read and reviewed the final version of the manuscript.

*Competing interests.* The authors declare that they have no conflict of interest.


*Acknowledgements.* Our special thanks go to Félix Pellerin, Maike Scheffold and Laurin Steidle for their valuable comments on the early version of this manuscript. This work was supported by the Center for Earth System Research and Sustainability (CEN), University of Hamburg, and contributes to the Cluster of Excellence "CLICCS - Climate, Climatic Change, and Society".



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



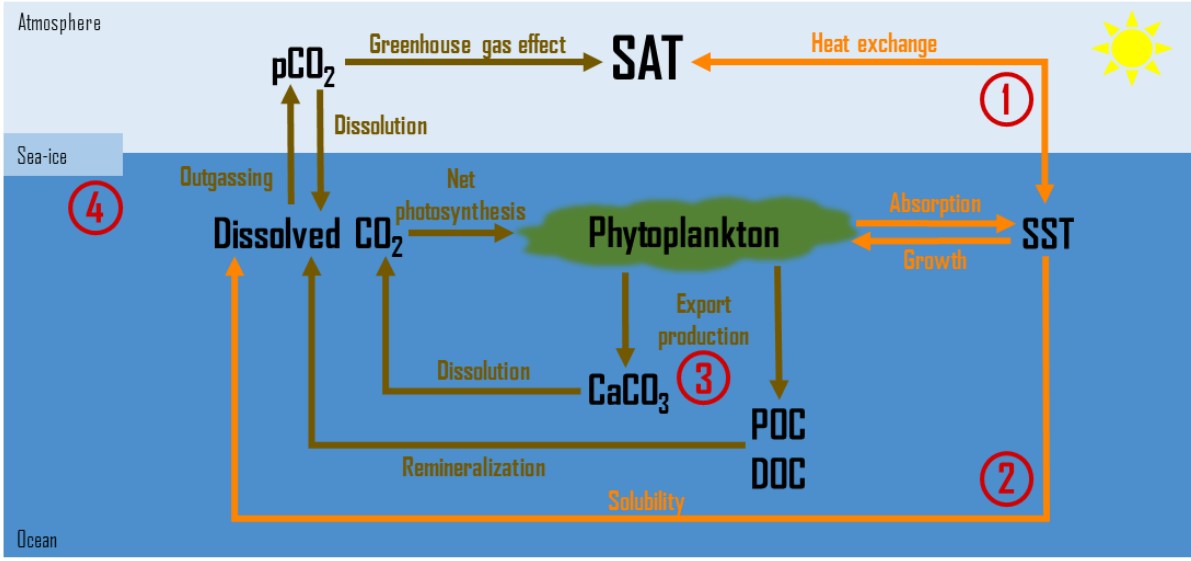

**Figure 1.** Representation of the four different biologically-induced pathways that affect the atmospheric temperature. (1) Marine biota via phytoplankton light absorption increases the SST, changing therefore the air-sea heat exchange and the atmospheric temperature. (2) Changes in SST also alter the solubility of $CO_2$ and its dissolved concentration. In turn, changes in dissolved $CO_2$ concentrations alter the air-sea $CO_2$ exchange and thus the greenhouse gas effect. (3) Phytoplankton light absorption modifies the marine biogeochemical cycles and particularly the export production of carbon. These changes in export production of carbon modify the dissolved $CO_2$ concentration and the greenhouse gas effect. (4) A warmer surface of the ocean can decrease the sea-ice extent. A reduction of sea-ice cover increases the air-sea $CO_2$ exchange area, changing the greenhouse gas concentrations. SAT = surface atmospheric temperature. SST = sea surface temperature. $CaCO_3$ = calcium carbonate. POC = particulate organic carbon. DOC = dissolved organic carbon.





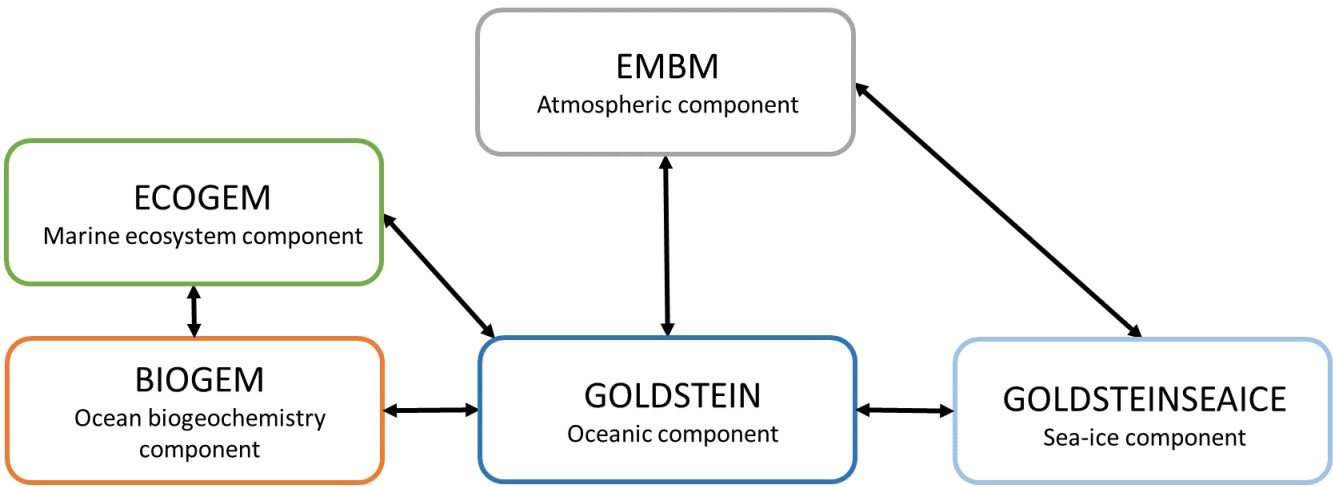

**Figure 2.** Representation of the components of the EcoGENIE model. The black arrows indicate the link between the different climatic components.





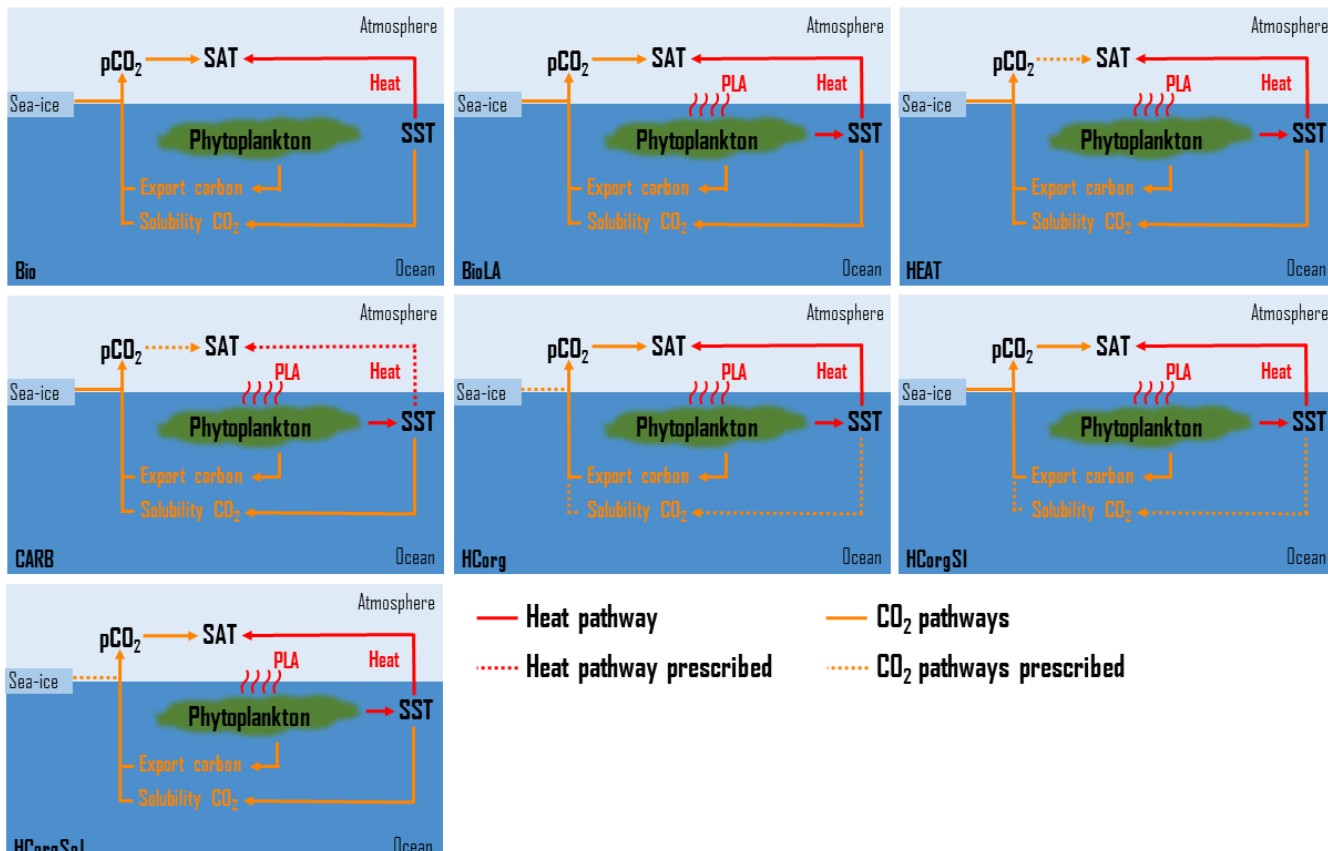

**Figure 3.** Sketch representing the climate pathways involved in the seven simulations conducted with EcoGENIE (PLA = Phytoplankton Light Absorption). Note that this figure is a simplification of Figure 1, only the relevant pathways are represented. The names of the simulations are on the bottom left of each panel. The dashed arrows indicate the climate pathways prescribed.







**Figure 4.** Global average of the different air-sea heat fluxes (W/m$^2$) for the seven simulations. (a) Net shortwave radiation at the top of the atmosphere. (b) Net re-emitted longwave radiation. The net longwave radiation is negative because it is dominated by the outgoing longwave radiation. (c) Sensible heat flux. (d) Latent heat flux. The color coding between the panels remains the same.





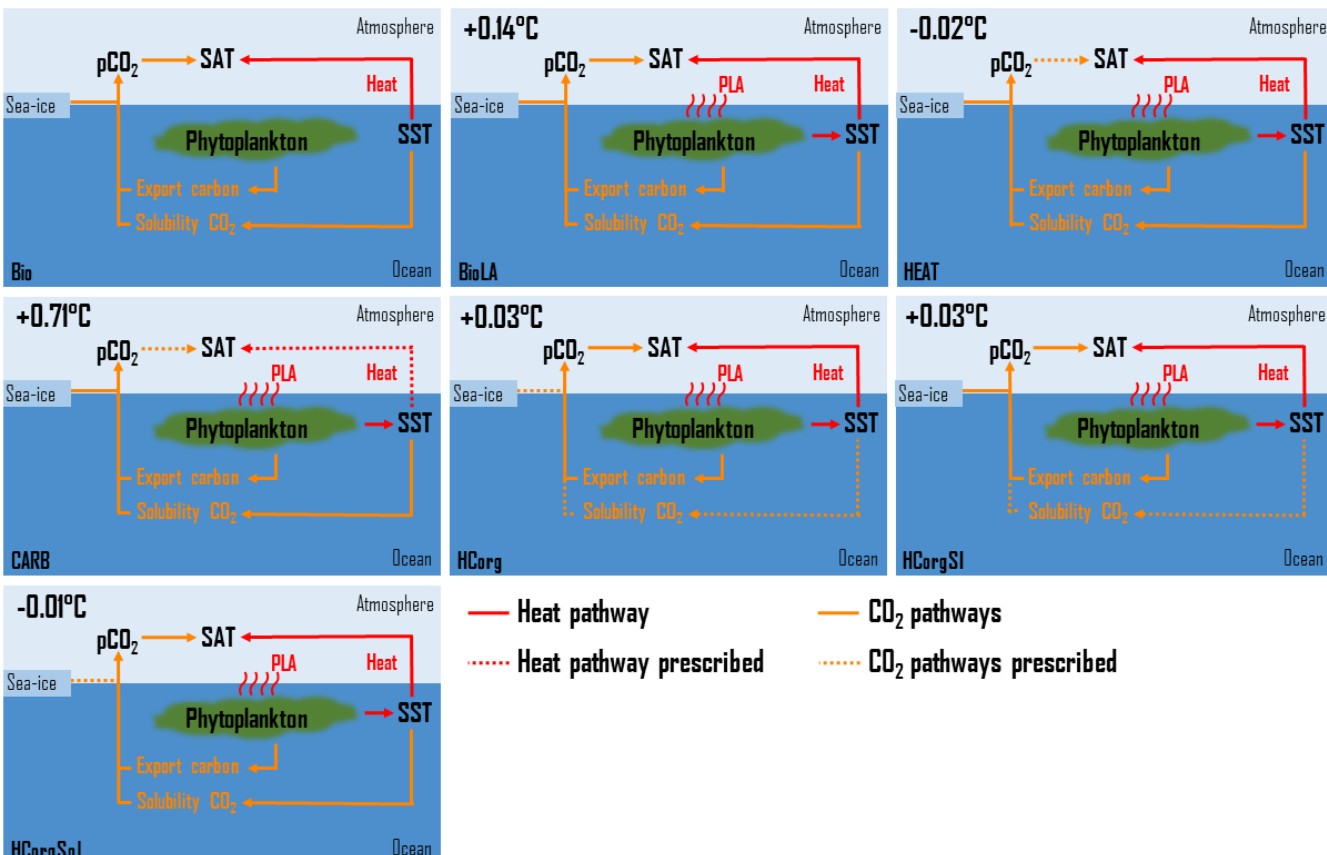

**Figure 5.** Sketch representing the surface atmospheric temperature (SAT) changes between the simulations and the reference run. On the top left corner is located the value of SAT change compared to *Bio*. The rest of the sketch is similar to Figure 3.





**Table 1.** Description of the simulations conducted with EcoGENIE. All the simulations consider phytoplankton light absorption except the reference run Bio.

| Name | Characteristics |
| --- | --- |
| Bio | Reference run |
| BioLA | Run with all pathways included |
| HEAT | Run with prescribed $CO_2$ pathway |
| CARB | Run with prescribed heat flux pathway |
| HCorg | Run with prescribed $CO_2$ solubility and sea-ice extent pathways |
| HCorgSI | Run with prescribed $CO_2$ solubility pathway |
| HCorgSol | Run with prescribed sea-ice extent pathway |

**Table 2.** Chlorophyll concentration ($mgChl/m^3$), sea and atmospheric surface temperature (°C) for the sensitivity analysis of the climate. The difference represents the value of Sensi320 minus the value of Sensi280.

| Simulation | Chloro. conc. ($mgChl/m^3$) | SST (°C) | SAT (°C) |
| --- | --- | --- | --- |
| Sensi280 | 0.1177 | 16.78 | 11.92 |
| Sensi320 | 0.1175 | 17.17 | 12.44 |
| Difference | -0.0002 | 0.39 | 0.52 |

**Table 3.** Changes in air-sea $CO_2$ exchange ($mol/m^2/yr$ and %) regarding the sensitivity of the system towards the interplay between $CO_2$ and heat. For the first sensitivity analysis, the SST is increased by 1°C while for the second analysis, the annual mean wind speed is raised by 0.2 m/s. The third row corresponds to the maximum difference of SST between the simulations.

| Sensitivity analysis | $F_{CO_2}$ ($mol/m^2/yr$) | Changes (%) |
| --- | --- | --- |
| +1°C | $+4.26 \cdot 10^{-5}$ | 2.58 |
| +0.2 m/s | $+1.44 \cdot 10^{-4}$ | 8.69 |
| +0.08°C | $+3.40 \cdot 10^{-6}$ | 0.21 |





**Table 4.** Sea surface temperature (°C) and surface chlorophyll biomass (mgChl/m$^3$). There is no value for the simulation CARB because we run the model with an uncoupled ocean-atmosphere setup.

| Simulation | SST (°C) | Chlorophyll biomass (mgChl/m$^3$) |
|---|---|---|
| Bio | 15.26 | 0.09949 |
| BioLA | 15.34 | 0.11178 |
| HEAT | 15.25 | 0.10827 |
| CARB | - | - |
| HCorg | 15.30 | 0.10964 |
| HCorgSI | 15.30 | 0.10964 |
| HCorgSol | 15.28 | 0.10891 |

**Table 5.** Comparison of the atmospheric $CO_2$ concentration (ppm) for the seven simulations.

| Simulation | Atmospheric $CO_2$ (ppm) |
|---|---|
| Bio | 169 |
| BioLA | 178 |
| HEAT | 171 |
| CARB | 178 |
| HCorg | 174 |
| HCorgSI | 174 |
| HCorgSol | 175 |

**Table 6.** Comparison of important atmospheric properties: specific humidity (g/kg) and evaporation (mm/yr) for the seven simulations.

| Simulation | Specific humidity (g/kg) | Evaporation (mm/yr) |
|---|---|---|
| Bio | 11.762 | 834.70 |
| BioLA | 11.818 | 835.65 |
| HEAT | 11.794 | 836.28 |
| CARB | 11.845 | 835.96 |
| HCorg | 11.814 | 835.54 |
| HCorgSI | 11.814 | 835.54 |
| HCorgSol | 11.818 | 835.65 |



**Table 7.** Global surface atmospheric temperature and changes compared to the reference simulation (°C). In the second column, a positive value indicates a higher while a negative value indicates a lower surface atmospheric temperature in the respective simulation compared to the reference simulation.

| Simulation | SAT (°C) | Changes (°C) |
|---|---|---|
| Bio | 9.31 | - |
| BioLA | 9.45 | +0.14 |
| HEAT | 9.29 | -0.02 |
| CARB | 10.02 | +0.71 |
| HCorg | 9.34 | +0.03 |
| HCorgSI | 9.34 | +0.03 |
| HCorgSol | 9.30 | -0.01 |





**Table A1.** Size of the different plankton functional types ($\mu$m) used during the simulations.

| PFT | Size ($\mu$m) |
| --- | --- |
| Phytoplankton | 46.25 |
| Zooplankton | 146.15 |

**Table B1.** Sea surface temperature (°C) and atmospheric $CO_2$ concentration (ppm) for simulations with and without a seasonal cycle.

| Simulation | SST (°C) | Atm. $CO_2$ conc. (ppm) |
| --- | --- | --- |
| Seasonal cycle | 16.11 | 184 |
| Non-seasonal cycle | 15.34 | 178 |

**Table C1.** Global sea-ice cover (%) and global sea-ice area (km$^2$) for the different simulations.

| Simulation | Sea-ice cover (%) | Sea-ice area (km$^2$) |
| --- | --- | --- |
| Bio | 9.79 | $3.60 \cdot 10^7$ |
| BioLA | 9.76 | $3.59 \cdot 10^7$ |
| HEAT | 9.91 | $3.64 \cdot 10^7$ |
| CARB | 8.60 | $3.16 \cdot 10^7$ |
| HCorg | 9.92 | $3.65 \cdot 10^7$ |
| HCorgSI | 9.92 | $3.65 \cdot 10^7$ |
| HCorgSol | 10.02 | $3.68 \cdot 10^7$ |

**Table D1.** Precipitation (mm/yr) for the different simulations.

| Simulation | Precipitation (mm/yr) |
| --- | --- |
| Bio | 834.62 |
| BioLA | 837.07 |
| HEAT | 836.30 |
| CARB | 834.05 |
| HCorg | 837.00 |
| HCorgSI | 837.00 |
| HCorgSol | 837.07 |