# Peer review of "Climate pathways behind phytoplankton-induced atmospheric warming"

_Biogeosciences, 2021_

## Referee Comment (RC2)

**Review of "*Climate pathways behind phytoplankton-induced atmospheric warming*" by Asselot et al. (2021) for Biogeosciences**

**Paper Synopsis:**

In this paper, Asselot et al. analyse the climatic impacts of including a representation of phytoplankton light absorption (PLA) in the Earth system model of intermediate complexity (EMIC) ecoGEnIE. By intercepting more light PLA results in greater surface water warming than would otherwise be simulated, which has knock-on effects for both heat and CO2 exchange across the air-sea boundary. However, the relative importance of these different climate pathways has not been analysed. To do this, Asselot et al. have implemented a PLA parameterisation they recently developed for ecoGEnIE (in: Asselot et al, 2021, GRL) to replace ecoGEnIE's original simplified light representation, and then rerun simulations keeping different climate impact pathways fixed in order to assess the importance of each for PLA. Based on this they find that sir-sea CO2 flux has a much larger effect on atmospheric temperature than the air-sea heat flux, making the former the dominant climate pathway for PLA.

**General Comments:**

In general this is an interesting and worthwhile paper that is suited to the scope of Biogeosciences (connecting ecology, biogeochemistry, and climate) and has a clear rationale (to disentangle the effects of PLA on climate), but would benefit from further explanation, clarifications, and details throughout in order to make the methodology and significance clearer. The manuscript could also do with a tidy up in some areas (e.g. paragraph formatting, confusing phrasings), but as this will be dealt with in copy-editing I have mostly focused on clarity and scientific content.

The Abstract could be more detailed on what has been done and why, for example outlining how the relative importance of heat and CO2 exchange has not previously been disentangled (and so provides justification for why this study is useful and timely), and directly mentioning in the abstract that you've implemented a new PLA parameterisation (as currently it just says you use ecoGEnIE without being clear on how it's different from previous usage). Some more detail on the results would be useful in the Abstract as well, for example specifying the direction and magnitude of each pathway (e.g. $+x^{o}C$, $-y^{o}C$), and mentioning here or earlier in the Abstract which specific mechanisms are involved for both air-sea CO2 and heat fluxes (e.g. warming from PLA reduces capacity to dissolve CO2, changes in latent heat, etc.). Conversely, the sentence on lines 6-8 on model configuration seems more methodological detail than necessary in an Abstract.

The Introduction could so with more detail on exactly how PLA affects each pathway – for example, how PLA impacts the biological pump and the carbon cycle, or the relation of sea ice with gas exchange. At the moment several mechanisms are not really explained until they come up in the Results, when ideally they'd be flagged and explained first in the Introduction for the benefit of readers less familiar with PLA or the mechanisms. I'd also like to see some more detail on how the PLA scheme differs from the original ECOGEM setup for those not familiar with how light is treated in these models. The experimental set-up is clear though and capable of disentangling the potential climate impact pathways.

In several places numbers from this study are qualitatively compared with previous studies, when a quantitative comparison would be more informative for the reader. Similarly, the reader is referred to Asselot et al. (2021) several times for explanations of particular mechanisms or methodological

choices when it'd be better to briefly summarise those explanations in this paper as a recap for readers who may not have read the former paper or recall all the details.

The authors make a reasonable case for why this study is still useful despite the lack of model seasonality (in contrast to their recent study Asselot et al. (2021)), but I feel that the statements that it has no effect on the results to be overstated given that it is clear that there are indeed some effects (reducing the net impact of including PLA from 0.45°C in Asselot et al. (2021) to 0.14°C here). A number of other methodological choices and discussion points, for example the surprisingly low atmospheric CO2 model baseline of ~170ppm, AMOC strength in HEAT, drivers of HCorg vs. Bio biomass differences, whether temperature-dependent remineralisation is enabled, and not using ECOGEM's size class functionality, could also do with more explanation or clarification. I'm also not convinced of the utility of the current sensitivity analyses – see my specific comments for details.

Lastly, some wider implications or significance should be signalled in the Conclusions, for example how these results might relate to current Earth system models' projections for climate change and if they might differ if they included PLA.

**Specific Comments:**

Line 6-8:    This sentence on model configuration seems more methodological detail than necessary in an abstract – the extra space can be used to give a little more rationale and results details.

Line 27:    Given that this model setup has no seasonality it's arguable whether you're able to analyse "varying magnitude" in this study – unless you mean ESMs more broadly than this study, or you mean the relative magnitudes between the different climate impact pathways? If the latter better to say relative rather than varying, as varying implies some analysis of model variability.

Line 31:    "an increase of SST between 0.5-2C" relative to what – presumably no PLA? Need to clarify what exactly these past results are showing.

Line 37:    Is the 0.5C increase global? Also, better to say 1C at end of line to keep consistent units.

Line 50:    Add "both" before "air-sea heat and CO2 exchange" for sentence clarity.

Line 57-58:    More detail on the effects of PLA on the biological pump in Section 1 would be useful, as at the moment it is not explained until the results. Similarly with sea ice – a brief explanation stating that sea ice acts as an ocean cap that blocks gas exchange (and so lower extent means more gas exchange can occur) would be useful in Section 1.

Line 75:    "the sensitivity of atmospheric CO2" to what (or do you mean long-term variability?), and how does the organic carbon pump explain it? Need a little more explanation if bringing up a past model usage.

Line 76-77:    A minor point, but the current norm with the model developers is to spell cGENIE/EcoGENIE as cGEnIE/ecoGEnIE with a small n (as it's G̲rid E̲n̲abled...), so you may want to match that for literature consistency (although not super important). More importantly, in the next line GENIE should be specifically **c**GEnIE, as the two models have diverged and ECOGEM has been specifically developed for cGEnIE rather than GENIE.

Line 102-122: A bit more detail for readers not familiar with cGEnIE/ecoGEnIE might be valuable here. For example, on how remineralisation is dealt with (important for biological pump impacts),

what ecological processes are temperature or size dependent (e.g. predation, nutrient uptake, photosynthetic rates), how CaCO3 is dealt with (alluded to in Figure 1 but not elsewhere – not so critical with no calcifier PFT available though). These may not be so relevant for this particular study, but unfamiliar readers might want extra context on how this particular model represents biogeochemistry.

Line 114: Presumably you do not activate temperature-dependent remineralisation as well (as it is not mentioned)? It is not a standard option in ecoGEnIE but is available, and some recent papers (Crichton et al, 2021, GMD; Armstrong McKay et al., 2021, ESD) showed how turning on TDR (as well as ECOGEM) in cGEnIE can significantly alter global export magnitude and patterns. Would therefore be useful to specify which version is being used, as PLA-induced warming would have different impacts on remineralisation depending on it.

Line 116: Based on Table A1 though you only use one size class per PFT in this study despite more being possible, justified in Appendix A by more size classes having less of an effect than PLA in Asselot et al. (2021) and for saving computational time. One size ECOGEM is better than BIOGEM even with only one size class for each (as you still get explicit OM, flexible stoichiometry, more temperature sensitivities, etc.), but you'd still miss out on some extra ecological dynamics in response to warmer surface water (i.e. the combined effect of PLA *and* ecological complexity). You found a relatively smaller impact of increased ecosystem complexity in Asselot et al. (2021) on atm. CO2 relative to PLA, but given the smaller magnitude changes in this study it might play a non-trivial role, with for example your '6P6ZLA' simulations in Asselot et al. (2021) showing a ~10ppm difference in atm. CO2 to '1P1ZLA' versus a max. ~9ppm difference from your reference simulation here. Of course your focus here is on disentangling the effects of PLA so it does make some sense to simplify other aspects to make those effects clearer (though it'd be good to more clearly state as such in the main text rather than in the Appendix), but if possible it'd make an interesting sensitivity analysis to repeat Bio & BioLA with multiple size classes and see how that affects the net PLA impact. Also, while reducing size classes does save some computational time, from my own experience it'd likely be on the order of hours per run on a standard core, so not entirely prohibitive.

Line 124: It would be useful in this subsection to briefly mention how your new light parametrisation differs from ecoGEnIE's original Ward et al. (2018) light scheme (or cGEnIE's) for readers who may be familiar with cGEnIE or ecoGEnIE but not Asselot et al. (2021), as currently only the new scheme is described. Of course one can go to your recent paper for the full details, but a quick recap would be helpful!

Line 125-127: This is a reasonable simplification and likely won't alter your overall qualitative findings as you state, but I'm not sure it can be said to not affect your results. Given you describe the significant impacts of short-lived seasonal algal blooms in the Introduction, it stands to reason that an annual mean approach will probably underestimate PLA's impact on production and surface water warming. Given technical limitations it's probably fair not to include this in this paper (though I am intrigued as to why – see next-but-one comment), but it's no doubt a limitation to this study and should be stated as such.

Line 168: It would be useful to briefly describe how PLA affects CO2 via these parameters here – presumably warmer surface water from activating PLA reduces solubility and sea ice fraction, but it's worth stating so explicitly for clarity.

Line 178: How come a seasonal SST cycle was possible in Asselot et al. (2021) but not in this paper (except Appendix B) if it's effectively the same model code otherwise? Judging by the PLA impact in Asselot et al. (2021) versus the BioLA vs. Bio numbers in this study it seems to have quite large impact on the total warming magnitude. While unlikely to change the qualitative importance of

each climate pathway in this study, it can't entirely be ruled out that larger peak warming with seasonality-enabled could make non-CO2 pathways relatively more impactful.

Line 190:     So it is exactly the same scheme (including ECOGEM plus your new PLA scheme), just with this parameter set to 0? Important to be clear on as would affect baseline inter-comparability if they were using different modules. If it is just that parameter then it should be fine.

Line 196:     169 ppm for default scenario atmospheric CO2 seems rather low – it effectively means you're looking at an LGM ocean. Looking at lines 175-178 & Asselot et al. (2021), you spin up BIOGEM with 278ppm constant and then restart with ECOGEM+PLA, implying that CO2 falls by ~110ppm during the first ~700 years of your experiments before stabilising. This doesn't occur in the original ECOGEM version, with ecoGEnIE set up to match quasi-modern observations, so must be a result of adding PLA or some other modification you've made to ECOGEM (perhaps allowing deeper PP?). I know you're primarily interested in the relative impact of PLA on different climate pathways rather than overall magnitude of climate impact or state, but it strikes me as an unusual baseline climate state to end up using (even for an EMIC!) versus a more usual pre-industrial Holocene baseline, and might make inter-comparison with other processes and model developments trickier. More importantly the relative importance of PLA's climate impact pathways might also be state-dependent and vary depending on baseline, which could affect your results. At the very least I think this should be flagged here, and it'd be useful to know why CO2 falls so much in response to modified ECOGEM.

Another similar issue is model-data comparison – if CO2 drifts this much, does adding PLA change ecoGEnIE's match with observations, or was this covered in Asselot et al. (2021)? Some supplementary plots to demonstrate this might be necessary if there are noticeable differences.

Line 215-218:  You're not so much analysing the model's climate variability here as the difference between two different climate baselines (climate variability would imply seasonality, multi-decadal oscillations, etc.). And how do these runs compare with the 169ppm baseline in the main default scenario? A ~110ppm difference would be more significant than the 40ppm discussed here, and given that your scenarios are all around ~170ppm (Table 5) I'm not sure why comparing just 280 and 320 ppm to each other is so useful here.

Line 220-223:  And how would this affect the main results, in particular BioLA vs. Bio? A sensitivity analysis would be most useful in determining how sensitive the overall pattern of findings are to certain parameters such as baseline pCO2, SST, wind speed, seasonality, etc. by repeating some or all of your main simulations with those different baselines, whereas simply showing that higher pCO2 leads to greater SAT & SST is somewhat unsurprising. If more detailed sensitivity analyse simulations were conducted (which ideally they should, or the impacts of critical factors explored and discussed in greater detail some other way) then the supplement could be used to avoid manuscript clutter with just a brief summary in the main text.

Line 230:     The change in FCO2 is indeed small, but on long timescales it could still affect the ocean carbon sink capacity quite a bit (after all, if SST had marginal impact on CO2 flux then there wouldn't be worries about the solubility pump and therefore the ocean carbon sink declining with global warming). However, in equilibrium runs with relatively stable SATs/SSTs as in this study this isn't so important, and the maximum SST difference from your main runs yields a far smaller change. Still, it might be beneficial to calculate how much a 0.21% increase in FCO2 would affect carbon reservoir size over the duration of your runs, if only to demonstrate it's only a minor factor.

Line 234-235: But what's the implication of the larger impact of changed wind speed baseline on FCO2 for your results then? At the moment you use the comparison to imply SST's effect is negligible, but don't go in to the implications of your wind speed sensitivity calculation.

Line 237-238: Additionally cGEnIE/ecoGEnIE is rather low resolution anyway and biota can't move between grid cells, so will not resolve some key regional processes even if seasonality was included.

Line 246:      The CARB simulation is not described in Section 5.1, presumably because it's not so interesting for examining ocean dynamics (as you mention in outlining the scenarios), but this could bear repeating here even if briefly (much like CARB is only briefly mentioned in 5.2.1 to state atmospheric CO2 is the same as BioLA).

Line 249-251: I don't think this particular mechanism (PLA increasing chlorophyll biomass via changes in OM export) has been described in this manuscript prior to this point, and as readers haven't necessarily read the cited papers it would be therefore useful to explain this process here in a little more detail. Is it because activating PLA brings production closer to the surface and therefore boosts surface nutrient recycling and surface biomass? Also, "shallower downward flux of OM" is slightly confusing phrasing.

Line 253-255: It would be useful to provide those previous estimates here for the reader to compare the current estimate with. Looking at Asselot et al. (2021), is the comparable number there 0.45C? If so it's quite a big difference, and emphasises the point earlier that no-seasonality may not affect the overall pattern of findings but will still affect your results to some degree (which you've usefully stated here).

Line 259-260: The impacts of stronger overturning fit with your results, but why is overturning a whole Sverdrup stronger in HEAT if the CO2->SAT forcing is the same? Needs some explanation, as a ~13% increase is quite a big difference.

Line 268-281: The differences between HCorg and HCorgSol solubility and PO4/chl/SST are explained in this paragraph, but not why the HCorgX scenarios in general have higher chl biomass & SST than the reference run. Presumably as with HCorg vs. HCorgSol it might be explained by higher PO4 in HCorg vs. Bio (which leads to higher biomass and therefore higher SST due to PLA), but an explanation of how these differences versus the default simulation arise (and indeed why PO4 is higher, explaining in more detail the role of the biological pump here) would be useful at the outset of this paragraph to guide the reader along. I'm also wondering to what extent using only one size class each for phyto- and zooplankton might affect these HCorgX simulations, given the role different size classes play in POM vs. DOM production.

Line 289:      Would be useful to state what the previous estimate of BioLA vs. Bio CO2 was to avoid the reader having to flick back to Asselot et al. (2021) to compare it themselves.

Line 295-297: What do you mean by more "important" here – higher, I guess? A bit unclear phrasing to me as it stands (similarly in final paragraph sentence.)

Line 298:      The remineralisation *rate* would only be higher if temperature-dependent remineralisation (TDR) is activated, otherwise remin. rates follow a fixed profile in cGEnIE & ecoGEnIE (see Crichton et al., 2021 & Armstrong McKay et al., 2021 for discussion of TDR in cGEniE/ecoGEnIE). If the former is the case in this study then you need to state this in the Methods, but if not then higher remin. rates can't be the cause of increased dissolved CO2 – instead it'd most likely be higher

*production* rates leading to more remin. despite remin. being at the same rate. I suspect this might be what you meant here anyway – if so, should be clear on terminology here to avoid confusion.

Line 397-398: You should state the mechanism for how PLA increases atmospheric CO2 here (i.e. PLA = surface warming = lower solubility = more atm. CO2) for clarity.

Line 332-333: Highest as in least negative, but arguably a tad confusing phrasing as Bio/CARB as plotted have the smallest (negative) bars. Might be better to rephrase sentence starting in line 331 as "A *more* negative value of net longwave heat flux indicates a greater loss of heat to outer space" along with next sentence accordingly.

Line 334: Refer the reader here to the next section for details on specific humidity (otherwise it feels like a skipped detail).

Line 339: This might be my more limited experience of climate thermodynamics relative to biogeochemistry here, but presumably in Bio the same amount of light enters the ocean but is absorbed less close to the surface (as no PLA), so activating PLA is not so much an "additional heat source" as heat absorbed closer to the surface in a way that affects the atmosphere on shorter timescales? If so then on longer ocean overturning and equilibration timescales could total ocean heat uptake in Bio eventually lead to higher outgoing radiation than in this spin-up?

Line 358: How much lower than previous values? Useful to state these things directly so the reader can easily see the difference.

Line 362: Confusingly phrased (had to re-read a few times to get it) – might be clearer rephrased as "*with lower humidity leading to higher evaporation rates*" or similar.

Line 374: As mentioned earlier, it's useful to specify what these previous estimates are for easy inter-comparison by the reader. In this case, if the relevant number from Asselot et al. (2021) is 0.45C then arguably in the next sentence ~25% of that value is not such a small difference.

Line 380-381: Presumably the global net longwave heat flux decreases because of the lower SST.

Line 412-417: I'd add that bringing in more plankton size classes (as well as more PFTs) would be useful for more complex ecological dynamics to emerge, which as well as enabling TDR could significantly affect the biological pump pathway. Additionally assessing if there's climate state-dependence, reflecting my earlier comments about the relatively low CO2 baseline. These could of course make nice sensitivity analyses for this present study, but if that's not possible in the time available then they'd certainly be useful next steps (and arguably increasing ecological complexity is a priority ahead of increasing atmospheric complexity with PLASIM). I'd also be interested how this would affect a transient warming simulation – would it likely amplify or dampen global warming? This could be something to draw out as wider implications too.

Figure 3: The simulation names might do with being bigger (and/or in top-left corners), as I didn't notice them at first. Also, currently the dotted lines initially seemed to imply that the prescribed pathways are the same in each case, whereas for example the pCO2->SAT prescription in HEAT is from Bio and in CARB is from BioLA. Maybe could add a label above prescribed lines indicating from which setup the prescription is coming from? It'd make the figure busier, but would also make it more self-standing without needing to cross-reference to the text so much.

*Dr. David A. McKay*

*27/8/2021*

**References:**

K. A. Crichton, J. D. Wilson, A. Ridgwell, P. N. Pearson, Calibration of temperature-dependent ocean microbial processes in the cGENIE.muffin (v0.9.13) Earth system model. Geosci. Model Dev. 14, 125–149 (2021).

D. I. Armstrong McKay, S. E. Cornell, K. Richardson, J. Rockström, Resolving ecological feedbacks on the ocean carbon sink in Earth system models. Earth Syst. Dyn. 12, 797–818 (2021).

---

## Referee Comment (RC3)

Alexandre Pohl (Biogéosciences, Dijon, FRANCE)
02 September 2021

**Summary:**

Asselot et al. study the mechanisms by which light absorption by the phytoplankton impacts the ocean-atmosphere system, global temperatures in particular. They use the Earth System Model of intermediate complexity GENIE, extended to include light absorption by Asselot et al. (2021), and conduct several numerical experiments by turning on and off key feedbacks. This approach allows the authors to quantify the impact of the different mechanisms studied. Asselot et al. conclude that the air-sea $CO_2$ exchange has a much larger impact on biologically-induced global climate warming than changes in the heat fluxes.

**General evaluation:**

I think that the authors approach an interesting topic. They present a clear modeling strategy. Most of the main text is clear and concise. Figures and Tables are mostly well adapted and satisfactorily convey the authors' message. However, several points should me made clearer. More importantly, I'm worried about key choices made by the authors regarding their modeling setup, which I think are not really obvious and have the potential to significantly impact the results and conclusions.

**Main comments:**

- Modeling setup:
  - Model spinup and equilibrium: Based on Section 3, the authors ran cGENIE with biogem for 10 kyrs and then restarted the model with ecogem for another 1000 years. First, I don't understand how ecogem can be run based on a biogem-only restart, but this is a technical point. More importantly, I don't think that this setup can lead to robust results. Indeed, changing from biogem to ecogem is expected to lead to important changes. An example is the pCO2 that drops from 278 ppm in the biogem run to 169 ppm in the Bio simulation. Such drastic change is expected to impact global climate and I don't think that 1000 years are enough to reach a new equilibrium. I would instead suggest running ecogem simulations for 10–20 kyrs and make sure that equilibrium is reached. Ensuring a robust equilibrium is particularly important considering the subtle changes reported between the different experiments (e.g., Table 7).
  - Climate state- and model configuration- dependence of the results: I also don't understand the choice of a cold climate (169 to 178 ppm; Table 5). I do expect the results of the study to be climate state-dependent. It would be interesting to determine if the same conclusions can be reached when using a higher baseline CO2 level (e.g., 350 ppm or above) (which might lead to very different results, due for instance to the lower sea-ice cover). At least, the authors should state that their results are expected to be climate state-dependent. I also think that it should be made clear that the present-day continental configuration is used. I also expect results to potentially vary with the land-sea mask.
  - Absence of seasonal cycle: Based on Appendix B, it appears that neglecting the seasonal cycle in the version of ecoGENIE modified to include light absorption leads to a global temperature difference bias (0.33 – 0.14 = 0.19 °C) that is larger than most temperature differences computed in Table 7. I think that the absence of seasonal cycle thus constitutes a major limitation to this work and would encourage the authors to repeat the experiments with seasonality.
  - Absence of size classes: I don't understand why the authors modified the model of Ward et al. In Appendix A, it is stated that it permits reducing computational time. However, a 10-kyr ecogem model run using the 36x36 grid with 16 vertical levels requires less than 5 days on a single core. Although I understand that using 32 levels probably makes the model more expensive, I guess that the model remains very fast to run and tractable. In

any way, please make sure to describe the model in a consistent manner. For now, it is not very clear: in section 2.1.3, the model of Ward et al. is described as including 2 PFTs with size classes, while in Appendix A, it is described as including 16 PFTs. There is a confusion between the number of plankton types (zoo vs. phyto) and the number of size classes.

- o Temperature-dependent remineralization: Is any temperature-dependent remineralization scheme employed in this study? It is not stated anywhere, but lines 249–251 suggest that the higher SSTs lead to a shallower remineralization. If so, please clarify this point and provide a reference.
- o Absence of light limitation by sea ice in ecogem: Although it probably has a minor impact, I also note that the attenuation of the photosynthetically available radiation by sea-ice in ecogem, as now part of the muffingen release, does not seem to be used in these experiments.
- o Absence of dynamical atmosphere: Based on Section 4.2, it seems that changing wind stress could have a major impact on the results. cGENIE wind fields are boundary conditions and do not vary with changing climate. I agree with the authors that a dynamical atmosphere would be useful (lines 412–413) but would rather present this as a current limitation / necessary next step rather than a possible way forward and encourage the authors to expand on this point.

- Model description: In section 2.1.2, the authors should clearly state that the description only refers to their model setup / the choices that they made. For instance, all productivity schemes of cGENIE do not include light nor iron limitation.

- "Previous estimates": Please provide the correspond values, on lines 252, 253, 289, 374.

**Other (mostly minor) points:**

- Line 8: "dissolved CO2" > "air-sea CO2 flux"?
- Line 59: "due to fluctuations"
- Line 62: "as follows"
- Line 71: "composed"
- Line 77: "cGENIE"
- Line 120: state variables for iron, too?
- Line 139: "(Eq. 2)"
- Lines 142–144: I don't follow this. It is stated that "the whole light absorption leads to heating of the water". Shouldn't the fraction used by the plankton be subtracted, according to line 142?
- Line 148: "(Eq. 3)"
- Line 152: delete "certain"?
- Line 162: "(Eq. 4)"
- Lines 200–201: I thought that atmospheric pCO2 was prescribed, based on lines 199–200?
- Section 4.1: I don't understand the utility of this section (which should be called "climate sensitivity" by the way?). Delete?
- Line 216: "differs"
- Line 227: "analyses"
- Line 228: "do not exceed the maximum difference of SST between our simulation results". I don't understand this.
- Line 230: "Even larger SST fluctuations … interface". Not shown?
- Lines 231–232: "We increase the wind speed by 0.2 m/s… (Knutson and Tuleya, 2004)." Please expand. It would be useful to the reader.
- Line 270: I would avoid calling chlorophyll concentration a "climate variable".

- Line 277: Is this difference larger than the one obtained when running the same experiment twice?
- Line 296: "biomass is more important in HEAT than in the reference simulation (Table 4)"
- Lines 308–310: please rephrase or add punctuation.
- Line 331: "a higher negative value" > "lower absolute value" may be clearer?
- Lines 337, 368: please delete "rather".
- Lines 351: "in these simulations"?
- Line 357:  "Oschlies (2004) and Lengaigne et al. (2009)"?
- Lines 364–365: "specific humidity and evaporation are higher in the simulation CARB than in BioLA"
- Line 371: please delete "indubitably".
- Line 401: formatting of the references.
- Lines 413–415: "Observations… […] Wurl et al. (2018)." Please delete or expand. As such, the message is not easy to understand.
- Appendix B1: should refer to Table B1.
- Line 435: "decreases the mean annual SST"
- Line 437: "dampens"
- Lines 444–445: "maximum global sea-ice cover change (or difference)"
- Caption of Fig. 2: EMBM should be defined.
- Figure 5: Unless I missed something, this figure can be deleted since it is redundant with Fig 3 and the 3rd column of Table 7.
- Table 1: "Bio – Reference run without phytoplankton light absorption".

---

## Author Comment (AC1)

Review of "Climate pathways behind phytoplankton-induced atmospheric warming" by Asselot et al.

We would like to thank the referee for the very thoughtful and constructive comments.

In this study, the authors explore the sensitivity of globally-annually-averaged atmospheric CO2 concentrations to phytoplankton induced surface ocean heating. Phytoplankton heat the surface ocean by absorbing radiation, which then heats the surrounding seawater and leads to changes in heat and carbon transfer with the atmosphere. Predicting even the sign of the change can be difficult since multiple physical factors (including changes to circulation) and chemical factors (changes to solubility) are relevant. The authors disentangle the relative strength of phytoplankton-mediated heat versus carbon transfer for influencing atmospheric CO2 via a series of idealized experiments in which physical and chemical factors are controlled. They find that phytoplankton heat absorption has a stronger influence on carbon exchange than heat exchange.

The study promises to be an excellent contribution to the question of how important the inclusion of phytoplankton heat absorption in climate models might be. Adding new code to existing models requires effort, and this study suggests this functionality alters heat and carbon fluxes and atmospheric CO2 concentrations. In their idealized framework, the effect on atmospheric CO2 is only 9 ppm on global annual average, suggesting phytoplankton heat absorption is a minor contributor. However, as the authors point out, this estimate may be a lower bound in less idealized conditions or the real world.

The experimental setup is thoughtfully constructed. Overall, results are discussed appropriately, although I found some critical information to be missing both from the description of the experimental setup as well as in the analysis of the model results. The paper is presented clearly, though I would like to see a modest expansion of the conclusions section to bring together the various experiments (including the sensitivity tests) in a more meaningful way.
I have some specific comments that should be addressed to improve the clarity of the manuscript:

- Figure 1 shows that the carbonate counter-pump is simulated, but the details of this and its implications and effects on the results are never discussed. Please provide the reader with more detail on this aspect of the biophysical feedback on carbon exchange.

The production and export of $CaCO_3$ in the surface of the ocean is linked to the export POM via a spatially uniform value which is modified by a thermodynamically-based relationship with the calcite saturation rate. The dissolution of $CaCO_3$ is treated the same way to that of the remineralization of POM (Ridgwell et al., 2007). In our previous study (Asselot et al., 2021), we show that phytoplankton light absorption accelerates the remineralization of POC at the surface. Because the dissolution of $CaCO_3$ and the remineralization of POM are treated the same way, phytoplankton light absorption also accelerates the dissolution of $CaCO_3$ at the surface. The accelerated dissolution of $CaCO_3$ and the accelerated remineralization of POM lead to a shallower downward flux of organic matter. However, our previous study evidences that the shallower downward flux of organic matter explain only a small part of the climate system's response (Asselot et al., 2021). Therefore the carbonate counter-pump has a small implication on our results.

- Depending on model formulation, phytoplankton-mediated heating of the surface ocean could have an effect on the top-down control of zooplankton on primary production. Please discuss if and how this might affect results.

With our model setup, due to phytoplankton light absorption the surface phosphate concentration increases by 15.15%. The chlorophyll biomass increases by 12.36% while the zooplankton biomass increases by 12.83%. The increase in surface phosphate concentration is larger than the increase in chlorophyll biomass while the increases in chlorophyll and zooplankton biomasses are similar.

These results indicate that the top-down control of zooplankton on phytoplankton limits the increase of chlorophyll biomass and is dominant compared to the bottom-up control of nutrients. A sentence has been added to the manuscript.

- The absence of nitrogen cycling is not discussed but could have additional consequences not modelled here. If phytoplankton warm a low-oxygen region, and this causes additional oxygen consumption, then there might be additional denitrification. More denitrification would lead to more nitrogen fixation downstream, which might increase biomass regionally (a change in the spatial pattern of NPP, which affects the overlap of solubility vs biomass) and therefore any pathway sensitivity of atmospheric CO2.

We would like to indicate that even without a nitrogen cycle, the main patterns of chlorophyll biomass are well represented in our simulations (figures below). The model underestimates the magnitude of chlorophyll biomass in the upwelling regions and polar regions. We agree and including a nitrogen cycle might increases the chlorophyll biomass in low-oxygen regions which might increase the pathway sensitivity of atmospheric $CO_2$. These speculations have been added to the conclusion part.

[Figure]

- How is wind stress forcing treated in the model? If it is like Weaver et al. (2001) then there is a change in the wind stress with a change in global temperature. I could not find this information in the manuscript and it has important implications for the results.

The wind forcing in the model is treated differently as in Weaver et al. (2001). The current model setup uses an identical prescribed wind field for all simulations. The wind stress cannot change if the global temperature changes. These sentences have been added to the manuscript.

- What are the temperature dependencies in BIOGEM/ECOGEM? Is there an approximation of a microbial loop? Is remineralization accelerated by warming? I could not find this information. This is related also to point (2).

In the model, there is an approximation of the near-surface loop or "microbial loop" of carbon cycling. The heating of the ocean doesn't directly accelerate the remineralization rate; the remineralization is not temperature-dependent. However, phytoplankton light absorption affects the physical properties of the ocean (e.g. enhanced upward vertical velocity, deeper MLD) and the ecosystem structure (e.g. increases in chlorophyll and zooplankton biomasses) and thus increases the remineralization rate.

- L175-181: Doesn't the application of ECOGEM change the biogeochemical distributions in the model? If so, 1,000 additional years of spin-up might not be enough. Are the sensitivity tests applied after the 10,000 years + 1,000 years, or are they applied after 10,000 years (is the atmospheric CO2 prescribed for the 1,000 year ECOGEM spin-up)? Is atmospheric CO2 allowed to stabilize in all model simulations, or are all simulations only run 1,000 years?

More details on experimental setup would be useful here.

Including ECOGEM changes, indeed, the biogeochemical properties of the ocean. Due to the single layer atmospheric component, the non-seasonality and the non-representation of the land dynamics, running the simulations for 1,000 years is sufficient to achieve steady-state. Actually, the steady-state is reached after 800 years of simulations. After the 1,000 years simulations, the atmospheric $CO_2$ concentrations are stabilized. The atmospheric $CO_2$ concentrations only vary of 0.05% every decade in the last century of the simulations (see figure below). The sensitivity analyses are conducted after the 10,000 years + 1,000 years model runs. During the simulations (1,000 years long model run) the atmospheric $CO_2$ concentrations are not prescribed except for the simulation CARB because in this simulation the ocean and the atmosphere are not coupled.

[Figure]

- It is not clear whether Fe is a prognostic variable. If it is, then are there temperature effects on Fe solubility (and therefore, bioavailability)?

In the model, iron is a prognostic variable. Currently, there is no temperature dependence either on Fe solubility in dust or Fe scavenging from the water column.

- Section 2.3: Does sea ice have no influence on heat exchange? Please explain.

Heat is exchanged between the atmosphere, the ocean and sea-ice. In the section 2.3 we only detail the total heat flux (ocean + sea-ice) going into the atmosphere. Figure 4 gives an overview of the total heat flux going into the atmosphere for the different simulations. However, our results indicate that sea-ice have a small effect on the heat exchange. The heat exchange is mainly influenced by the ocean.

- Table 2 and Section 4.1. I don't see these as being very important to the main message of the paper and suggest moving them into the Appendix. However, it is interesting comparing Tables 2 & B1. Inclusion of seasonal cycling has more of an effect on SST than 40 ppm change in CO2! What is the effect of seasonal cycling on chlorophyll?

We moved Table 2 and Section 4.1 in the Appendix.
Including a seasonal cycle increases the chlorophyll biomass by 0.039 mgChl/m$^3$. This is due to the warmer ocean, favoring the growth of phytoplankton.

- Section 4.2 could also move to an Appendix. But, as mentioned elsewhere- is the wind forcing different across the main model experiments due to differences in SST anomaly from pre-industrial state?

Section 4.2 has been moved to the Appendix. The wind forcing is similar between all the simulations. It cannot be affected by the changes in SST or atmospheric temperature.

- Conclusions are missing some wider speculation as well as more discussion of model limitations. What would the authors expect if a land model were to be included in their simulations? What about sea ice influencing heat flux, or carbonate counter-pump effects or changes to micronutrient availability? Feedbacks in a transient state?

We speculate that including a land model will still lead to an increase in atmospheric temperature due to phytoplankton light absorption but the magnitude of changes might be smaller. This is mainly due to the uptake of $CO_2$ by the vegetation, decreasing the atmospheric $CO_2$ concentrations and thus resulting in smaller increase in atmospheric temperature. If a land model were to be included, the magnitude of changes reported would be smaller but the sign would stay the same.

The sea-ice can influence the heat flux going into the atmosphere (see Equation 3). Figure 4 shows the total heat flux (ocean + sea-ice) going into the atmosphere. These details are added to the manuscript. However, our simulations indicate that changes in sea-ice don't play an important role in changes in air-sea $CO_2$ and heat fluxes.

The dissolution of $CaCO_3$ at the surface is accelerated by phytoplankton light absorption. Yet, our previous study (Asselot et al., 2021) indicates that this process explains only a small part of the climate system's response. The carbonate counter-pump does not play a major role to explain our results. This explanation is added to section 5.2.1.

The model does not include any temperature effects on the Fe solubility. However, here, we speculate on the consequences of implementing a temperature-dependency of iron solubility. According to previous experiments, the solubility of iron decreases when the oceanic temperature increases (Liu and Millero, 2002). Phytoplankton light absorption increasing the oceanic temperature might therefore reduce the Fe solubility and therefore its bioavailability. As a consequence, the limitation of phytoplankton growth by iron would increase, reducing the greater chlorophyll biomass with to phytoplankton light absorption.

Finally, I have some minor suggestions for language:

L1: "in which ways", or "ways in which"?
Changed

L10: "…the freely evolving solubility of CO2…" (due to what? Is it going up or down?)
When the solubility of CO2 can evolve freely, the atmospheric temperature slightly decreases due to the slight decrease in air-sea heat flux.

L11: Some kind of summary sentence that gives the results context would be useful here.
Added

L20: "evidence supports"
Changed

L29: "Models of differing complexity…"
Changed

L61: "…as follows…"
Changed

L71: "…composed of…"
Changed

L75: "…the sensitivity of atmospheric CO2 is mainly explained…"
Changed

L119: remove "availability"
Changed

L119: "prey"
Changed

L119: Table A1 shows that phytoplankton are ~3X smaller than the zooplankton. Does this mean there is no zooplankton grazing in the model?
Zooplankton grazing is represented in the model, even if phytoplankton are ~3X smaller than zooplankton.

L125: "For simplicity…"
Changed

Eqn 1: Does sea ice not affect light attenuation? Why not? Does this mean there is biomass under sea ice?
In the model, sea-ice affects the heat fluxes but for simplification, it does not affect the light attenuation. As a consequence, phytoplankton can grow in grid cells covered by sea-ice.

L132: "…total chlorophyll concentration .."
Changed

L142: "…is released in the form of…"
Changed

L151: "…received…"
Changed

Figure 1: There is no arrow between sea ice and anything else
We add the relevant arrow for our study. The arrow between sea-ice and outgassing of $CO_2$ is added to the figure.

Figure 5: Shouldn't "Bio" look more like Figure 1 (with an arrow going from SAT to SST?) Plus, CO2 should be able to enter the ocean in this simulation? Same for BioLA.
For simplification, on Figure 5 we only kept the arrow that go to the atmospheric temperature. Figure 5 only shows how the atmospheric temperature can be affected by the $CO_2$ and SST.

L203: Is this only the biological pump? What about CaCO3?
We revise or sentence and changed "biological pump" by "biogeochemical pumps" to include the soft-tissue pump plus the carbonate pump. Thus $CaCO_3$ is included in the biogeochemical pumps.

L216: "…concentrations differ."
Changed

L225: "…we ensure that the heat and CO2 interaction is negligible by …"
Changed

L227: "…analyses…"
Changed

L237: split into 2 sentences
Changed

L245: "Finally, the response of the surface atmospheric temperature due to changes in oceanic and atmospheric properties is studied"
Changed

L250: What is the temperature dependency that produces the shallower flux of OM due to warming from phytoplankton light absorption? (For those not familiar with your model).
We briefly summarize why the downward flux of organic matter is shallower with phytoplankton light absorption. This is due to: (1) A deeper mixed layer and therefore a more important mixing in the surface of the ocean (2) Enhanced upward vertical velocity, trapping more nutrients and organic matter at the surface of the ocean.

L251 (and elsewhere): "The chlorophyll biomass difference…" (Also applies to SST)
Changed

L264: remove "more important"
Changed

L 270: might this be due to the lack of an influence of sea ice on heat flux?
We added a sentence in the manuscript.

L 275/76: Is there a difference between HCorgSol and HCorg w.r.t. CaCO3 production?
The export production of CaCO3 is higher in HCorg compared to HCorgSol. This result does not change the main message of this paragraph because the $CO_2$-solubility is prescribed in the simulation HCorg, therefore the $CaCO_3$ production cannot affect the $CO_2$-solubility.

L 296/97/300: I think "more important" is not what is meant. "Larger" or "Greater"?
Changed

L 307: "…which increases…"
Changed

L 308-310: split into 2 sentences
Changed
L319: "…heat flux, explaining…"
Changed

L330: replace "pointing out" with "which indicates"
Changed

L332: "to outer space"
Changed

L362: "…, where the lower the humidity the higher the evaporation rate."
Changed

L371: remove "indubitably"
Changed

L375: maybe not "clearly", since I still have some questions about experimental setup and model assumptions beyond seasonality.
Changed

L388: "…identical responses…"

Changed

L 390: "…smaller increase…"
Changed

**Literature**

Asselot R, Lunkeit F, Holden PB, Hense I. The relative importance of phytoplankton light absorption and ecosystem complexity in an Earth system model. *Journal of Advances in Modeling Earth Systems*. 2021;13(5):e2020MS002110.

Liu X, Millero FJ. The solubility of iron in seawater. Marine Chemistry. 2002 Jan 1;77(1):43-54.

Ridgwell, A., Hargreaves, J. C., Edwards, N. R., Annan, J. D., Lenton, T. M., Marsh, R., Yool, A., and Watson, A.: Marine geochemical data assimilation in an efficient Earth System Model of global biogeochemical cycling, *Biogeosciences*, 4, 87–104, 2007.

Ward BA, Dutkiewicz S, Jahn O, Follows MJ. A size-structured food-web model for the global ocean. *Limnology and Oceanography*. 2012; 57(6):1877-91.

Weaver, Andrew J., et al. "The UVic Earth System Climate Model: Model description, climatology, and applications to past, present and future climates." Atmosphere-Ocean 39.4 (2001): 361-428.

---

## Author Comment (AC2)

**Review of "Climate pathways behind phytoplankton-induced atmospheric warming" by Asselot et al. (2021) for Biogeosciences**

**Paper Synopsis:**

In this paper, Asselot et al. analyse the climatic impacts of including a representation of phytoplankton light absorption (PLA) in the Earth system model of intermediate complexity (EMIC) ecoGEnIE. By intercepting more light PLA results in greater surface water warming than would otherwise be simulated, which has knock-on effects for both heat and CO2 exchange across the air-sea boundary. However, the relative importance of these different climate pathways has not been analysed. To do this, Asselot et al. have implemented a PLA parameterisation they recently developed for ecoGEnIE (in: Asselot et al, 2021, GRL) to replace ecoGEnIE's original simplified light representation, and then rerun simulations keeping different climate impact pathways fixed in order to assess the importance of each for PLA. Based on this they find that sir-sea CO2 flux has a much larger effect on atmospheric temperature than the air-sea heat flux, making the former the dominant climate pathway for PLA.

We would like to thank David Armstrong McKay for his very detailed, very helpful and supporting review.

**General Comments:**

In general this is an interesting and worthwhile paper that is suited to the scope of Biogeosciences (connecting ecology, biogeochemistry, and climate) and has a clear rationale (to disentangle the effects of PLA on climate), but would benefit from further explanation, clarifications, and details throughout in order to make the methodology and significance clearer. The manuscript could also do with a tidy up in some areas (e.g. paragraph formatting, confusing phrasings), but as this will be dealt with in copy-editing I have mostly focused on clarity and scientific content.

We will add more explanations, clarifications and details to make the manuscript clearer. We will also remove some tables and add figures where needed.

The Abstract could be more detailed on what has been done and why, for example outlining how the relative importance of heat and CO2 exchange has not previously been disentangled (and so provides justification for why this study is useful and timely), and directly mentioning in the abstract that you've implemented a new PLA parameterisation (as currently it just says you use ecoGEnIE without being clear on how it's different from previous usage). Some more detail on the results would be useful in the Abstract as well, for example specifying the direction and magnitude of each pathway (e.g. +xoC, -yoC), and mentioning here or earlier in the Abstract which specific mechanisms are involved for both air-sea CO2 and heat fluxes (e.g. warming from PLA reduces capacity to dissolve CO2, changes in latent heat, etc.). Conversely, the sentence on lines 6-8 on model configuration seems more methodological detail than necessary in an Abstract.

We will seize the suggestion and will revise the abstract including more specific details on the motivation, model design and results.

The Introduction could so with more detail on exactly how PLA affects each pathway – for example, how PLA impacts the biological pump and the carbon cycle, or the relation of sea ice with gas exchange. At the moment several mechanisms are not really explained until they come up in the Results, when ideally they'd be flagged and explained first in the Introduction for the benefit of readers less familiar with PLA or the mechanisms. I'd also like to see some more detail on how the PLA scheme differs from the original ECOGEM setup for those not familiar with how light is treated in these models. The experimental set-up is clear though and capable of disentangling the potential climate impact pathways.

We now admit that it is more comprehensible to introduce the pathways already in the introduction section and will do that in the revised version.

In several places numbers from this study are qualitatively compared with previous studies, when a quantitative comparison would be more informative for the reader. Similarly, the reader is referred to Asselot et al. (2021) several times for explanations of particular mechanisms or methodological choices when it'd be better to briefly summarise those explanations in this paper as a recap for readers who may not have read the former paper or recall all the details.

We agree and will add these estimates and briefly summarize the particular mechanisms and methodological choices in the revised manuscript.

The authors make a reasonable case for why this study is still useful despite the lack of model seasonality (in contrast to their recent study Asselot et al. (2021)), but I feel that the statements that it has no effect on the results to be overstated given that it is clear that there are indeed some effects (reducing the net impact of including PLA from 0.45oC in Asselot et al. (2021) to 0.14oC here). A number of other methodological choices and discussion points, for example the surprisingly low atmospheric $CO_2$ model baseline of ~170ppm, AMOC strength in HEAT, drivers of HCorg vs. Bio biomass differences, whether temperature-dependent remineralisation is enabled, and not using ECOGEM's size class functionality, could also do with more explanation or clarification. I'm also not convinced of the utility of the current sensitivity analyses – see my specific comments for details.
Lastly, some wider implications or significance should be signalled in the Conclusions, for example how these results might relate to current Earth system models' projections for climate change and if they might differ if they included PLA.

We would like to refer to the "specific comments" section. Furthermore, we already conducted additional sensitivity analyses to illustrate the effects of a more complex model with additional size classes.

**Specific Comments:**

Line 6-8: This sentence on model configuration seems more methodological detail than necessary in an abstract – the extra space can be used to give a little more rationale and results details.

Done

Line 27: Given that this model setup has no seasonality it's arguable whether you're able to analyse "varying magnitude" in this study – unless you mean ESMs more broadly than this study, or you mean the relative magnitudes between the different climate impact pathways? If the latter better to say relative rather than varying, as varying implies some analysis of model variability.

Indeed we referred to the pathways and thus replaced "varying magnitude" by "relative magnitude"

Line 31: "an increase of SST between 0.5-2C" relative to what – presumably no PLA? Need to clarify what exactly these past results are showing.

Yes, we refer to "no PLA"; we changed the sentence accordingly.

Line 37: Is the 0.5C increase global? Also, better to say 1C at end of line to keep consistent units.

The increase in atmospheric temperature is local. The units are changed.

Line 50: Add "both" before "air-sea heat and CO2 exchange" for sentence clarity.

Changed.

Line 57-58: More detail on the effects of PLA on the biological pump in Section 1 would be useful, as at the moment it is not explained until the results. Similarly with sea ice – a brief explanation stating that sea ice acts as an ocean cap that blocks gas exchange (and so lower extent means more gas exchange can occur) would be useful in Section 1.

In the introduction, more details will be provided on the different pathways, including the effects of PLA on the biological pump and consequences of sea ice on gas exchange.

Line 75: "the sensitivity of atmospheric CO2" to what (or do you mean long-term variability?), and how does the organic carbon pump explain it? Need a little more explanation if bringing up a past model usage.

We will revise this part.

Line 76-77: A minor point, but the current norm with the model developers is to spell cGENIE/EcoGENIE as cGEnIE/ecoGEnIE with a small n (as it's Grid Enabled...), so you may want to match that for literature consistency (although not super important). More importantly, in the next

line GENIE should be specifically cGEnIE, as the two models have diverged and ECOGEM has been specifically developed for cGEnIE rather than GENIE.

We agree and adopt the spelling cGEnIE/EcoGEnIE.

Line 102-122: A bit more detail for readers not familiar with cGEnIE/ecoGEnIE might be valuable here. For example, on how remineralisation is dealt with (important for biological pump impacts), what ecological processes are temperature or size dependent (e.g. predation, nutrient uptake, photosynthetic rates), how CaCO3 is dealt with (alluded to in Figure 1 but not elsewhere – not so critical with no calcifier PFT available though). These may not be so relevant for this particular study, but unfamiliar readers might want extra context on how this particular model represents biogeochemistry.

We will add more explanation: The surface export is divided between refractory organic matter remineralised close to the seafloor and labile organic matter remineralised in the upper water column. The remineralisation is not temperature-dependent in our model setup. Furthermore, because we do not consider a sediment component, all organic matter reaching the sea-floor is instantaneously remineralised.
The size-dependent ecophysiological parameters are: the maximum nutrient uptake rate, the cell carbon quotas, the grazing and the partitioning of dissolved and particulate organic matter.
The temperature-dependent parameters are: nutrient uptake, photosynthesis and predation.
Calcium carbonate ($CaCO_3$) is represented in the model and its dissolution below the surface is treated as the remineralization of particulate organic matter.

Line 114: Presumably you do not activate temperature-dependent remineralisation as well (as it is not mentioned)? It is not a standard option in ecoGEnIE but is available, and some recent papers (Crichton et al, 2021, GMD; Armstrong McKay et al., 2021, ESD) showed how turning on TDR (as well as ECOGEM) in cGEnIE can significantly alter global export magnitude and patterns. Would therefore be useful to specify which version is being used, as PLA-induced warming would have different impacts on remineralisation depending on it.

We thank the reviewer for pointing out these recent papers. In our model setup, we do not active the temperature-dependent remineralisation. A sentence is added in the manuscript.

Line 116: Based on Table A1 though you only use one size class per PFT in this study despite more being possible, justified in Appendix A by more size classes having less of an effect than PLA in Asselot et al. (2021) and for saving computational time. One size ECOGEM is better than BIOGEM even with only one size class for each (as you still get explicit OM, flexible stoichiometry, more temperature sensitivities, etc.), but you'd still miss out on some extra ecological dynamics in response to warmer surface water (i.e. the combined effect of PLA and ecological complexity). You found a relatively smaller impact of increased ecosystem complexity in Asselot et al. (2021) on atm. CO2 relative to PLA, but given the smaller magnitude changes in this study it might play a non-trivial role, with for example your '6P6ZLA' simulations in Asselot et al. (2021) showing a ~10ppm difference in atm. CO2 to '1P1ZLA' versus a max. ~9ppm difference from your reference simulation here. Of course your focus here is on disentangling the effects of PLA so it does make some sense to simplify

other aspects to make those effects clearer (though it'd be good to more clearly state as such in the main text rather than in the Appendix), but if possible it'd make an interesting sensitivity analysis to repeat Bio & BioLA with multiple size classes and see how that affects the net PLA impact. Also, while reducing size classes does save some computational time, from my own experience it'd likely be on the order of hours per run on a standard core, so not entirely prohibitive.

Though this is a bit out of scope of our study, where complexity is not the main issue, we repeated the simulations Bio and BioLA with 6 phytoplankton and 6 zooplankton species. The simulation without PLA is named Bio6 while the simulation with PLA is called BioLA6. The table below shows the quantities of the most important climate variables. The third row represents the differences between the simulations BioLA6 minus Bio6.

|  | Atm. CO2 conc. (ppm) | Chl. (mgChl/m3) | SST (°C) | SAT (°C) |
|---|---|---|---|---|
| Bio6 | 154 | 0.133 | 14.99 | 8.93 |
| BioLA6 | 159 | 0.140 | 15.04 | 8.97 |
| Difference | +5 | +0.007 | +0.05 | +0.04 |

Even with an increased ecosystem complexity, PLA increases the atmospheric temperature. However, the effect of PLA is reduced with a higher ecosystem complexity compared to the effect of PLA with a simple ecosystem community. This is due to the higher amount of carbon stored in the living biomass with increasing number of species, thus reducing the effect of PLA on the atmospheric CO2 concentration and on the climate system. We will add this information in the appendix.

Line 124: It would be useful in this subsection to briefly mention how your new light parametrisation differs from ecoGEnIE's original Ward et al. (2018) light scheme (or cGEnIE's) for readers who may be familiar with cGEnIE or ecoGEnIE but not Asselot et al. (2021), as currently only the new scheme is described. Of course one can go to your recent paper for the full details, but a quick recap would be helpful!

We added the sentence: "In the previous model version of Ward et al. (2018), light was only absorbed by phytoplankton. In our model version of Asselot et al. 2021, a new light scheme is implemented where the absorbed light by phytoplankton is converted into heat and affects the oceanic temperature. Furthermore light absorption takes place throughout the water column and is not restricted to the first oceanic layer."

Line 125-127: This is a reasonable simplification and likely won't alter your overall qualitative findings as you state, but I'm not sure it can be said to not affect your results. Given you describe the significant impacts of short-lived seasonal algal blooms in the Introduction, it stands to reason that an annual mean approach will probably underestimate PLA's impact on production and surface water warming. Given technical limitations it's probably fair not to include this in this paper (though I am intrigued as to why – see next-but-one comment), but it's no doubt a limitation to this study and should be stated as such.

We agree and will add this limitation in the discussion/conclusion section.

Line 168: It would be useful to briefly describe how PLA affects CO2 via these parameters here – presumably warmer surface water from activating PLA reduces solubility and sea ice fraction, but it's worth stating so explicitly for clarity.

We agree and will add a sentence accordingly.

Line 178: How come a seasonal SST cycle was possible in Asselot et al. (2021) but not in this paper (except Appendix B?) if it's effectively the same model code otherwise? Judging by the PLA impact in Asselot et al. (2021) versus the BioLA vs. Bio numbers in this study it seems to have quite large impact on the total warming magnitude. While unlikely to change the qualitative importance of each climate pathway in this study, it can't entirely be ruled out that larger peak warming with seasonality-enabled could make non-CO2 pathways relatively more impactful.

The seasonal cycle is removed for technical issues. In Asselot et al. (2021), we did not have to prescribe the SST in contrast to this study, at least for some simulations to look at specific pathways. The outputs of the SST field are annual means. Thus, it does not make sense to prescribe a yearly-averaged SST field while turning on the seasonal cycle. We decided to remove the seasonal cycle to be consistent with our prescribed non-seasonal SST field.
Enabling seasonality would lead to larger seasonal increase of temperature but it would also lead to larger seasonal decrease in CO2 solubility. Therefore, we don't think that the heat-pathway would overrule the CO2-pathway. We think that enabling seasonality would not change the qualitative importance of each climate pathway but we will add this point in the discussion section.

Line 190: So it is exactly the same scheme (including ECOGEM plus your new PLA scheme), just with this parameter set to 0? Important to be clear on as would affect baseline inter-comparability if they were using different modules. If it is just that parameter then it should be fine.

Yes, all the simulations include ECOGEM plus the new PLA scheme. And yes, only for our reference simulation Bio, the chlorophyll absorption coefficient is set to 0, thus chlorophyll does not absorb light at all.

Line 196: 169 ppm for default scenario atmospheric CO2 seems rather low – it effectively means you're looking at an LGM ocean. Looking at lines 175-178 & Asselot et al. (2021), you spin up BIOGEM with 278ppm constant and then restart with ECOGEM+PLA, implying that CO2 falls by ~110ppm during the first ~700 years of your experiments before stabilising. This doesn't occur in the original ECOGEM version, with ecoGEnIE set up to match quasi-modern observations, so must be a result of adding PLA or some other modification you've made to ECOGEM (perhaps allowing deeper PP?). I know you're primarily interested in the relative impact of PLA on different climate pathways rather than overall magnitude of climate impact or state, but it strikes me as an unusual baseline climate state to end up using (even for an EMIC!) versus a more usual pre-industrial Holocene baseline, and might make inter-comparison with other processes and model developments trickier. More importantly the relative importance of PLA's climate impact pathways might also be state-dependent and vary depending on baseline, which could affect your results. At the very least I think this should be flagged here, and it'd be useful to know why CO2 falls so much in response to modified ECOGEM.

Another similar issue is model-data comparison – if CO2 drifts this much, does adding PLA change ecoGEnIE's match with observations, or was this covered in Asselot et al. (2021)? Some supplementary plots to demonstrate this might be necessary if there are noticeable differences.

We are aware that an atmospheric CO2 concentration of 169 ppm for the reference scenario is indeed very low. Primary production takes place even in deeper layers of the water column and we also find deeper chlorophyll biomass. Our new model setup allows production until the sixth oceanic layer while the previous model setup only allows production in the surface layer. As a result, in our new model setup, more carbon is stored in the deeper ocean, reducing the atmospheric CO2 concentration.

Here we are interested in the relative impact of PLA on different climate pathways rather than on the overall magnitude of climate change. We agree that our quantitative estimates would be affected if the atmospheric CO2 concentration of the reference run would be higher but we assume that the qualitative estimates would be very similar and our conclusions will not change. If the atmospheric CO2 baseline is higher, also the heat budget would increase. As a consequence, the ocean would be warmer and the CO2 solubility would decrease, increasing the importance of the CO2 pathway in the phytoplankton-induced atmospheric warming.

Concerning the second point, in our previous paper (Asselot et al., 2021), we already compared our results with observations. We showed that the AMOC, the primary production, the export production of POC and the PO4 concentrations match observations. Here, for instance, we only compare the observed and modelled surface chlorophyll concentration.

[Figure]

The global pattern of surface chlorophyll biomass is in agreement with the satellite-derived estimates. The high latitudes show a large chlorophyll biomass while the subtropical gyres indicate a low chlorophyll biomass. However, the model underestimates the magnitude of the surface chlorophyll biomass. This is particularly true in the northern polar region and the upwelling regions. These limited agreements with observations are in line with the results of Ward et al. (2018).

Line 215-218: You're not so much analysing the model's climate variability here as the difference between two different climate baselines (climate variability would imply seasonality, multi-decadal oscillations, etc.). And how do these runs compare with the 169ppm baseline in the main default scenario? A ~110ppm difference would be more significant than the 40ppm discussed here, and given that your scenarios are all around ~170ppm (Table 5) I'm not sure why comparing just 280 and 320 ppm to each other is so useful here.

As also suggested by Reviewer 1, we moved the section 4.1 (climate variability) to the appendix in the revised manuscript. The following information is now available in the table where we compare the results of the reference (169ppm) and a sensitivity model run (280ppm).

| | Atm. CO2 conc. (ppm) | Chl. (mgChl/m3) | SST (°C) | SAT (°C) |
|---|---|---|---|---|
| Exp_1 | 169 | 0.09949 | 15.26 | 9.31 |
| Exp_2 | 280 | 0.1177 | 16.78 | 11.92 |
| Difference | -111 | -0.01821 | -1.52 | -2.61 |

The surface chlorophyll biomass, the SST and SAT are lower in our reference simulation (169ppm) compared to the sensitivity simulation (280ppm).

Line 220-223: And how would this affect the main results, in particular BioLA vs. Bio? A sensitivity analysis would be most useful in determining how sensitive the overall pattern of findings are to certain parameters such as baseline pCO2, SST, wind speed, seasonality, etc. by repeating some or all of your main simulations with those different baselines, whereas simply showing that higher pCO2 leads to greater SAT & SST is somewhat unsurprising. If more detailed sensitivity analyse simulations were conducted (which ideally they should, or the impacts of critical factors explored and discussed in greater detail some other way) then the supplement could be used to avoid manuscript clutter with just a brief summary in the main text.

We find that with an increase of 40 ppm, the changes in surface chlorophyll biomass are lower than the changes in surface chlorophyll biomass due to PLA, indicating that surface chlorophyll biomass is more "sensitive" to PLA than an increase of 40 ppm in pCO2. Yet, due to time constraints, we refrain running all experiments again with different parameters for a more detailed sensitivity study. We also think that our current findings are meaningful as it provides a first glimpse into different pathways even if we cannot answer all potential questions related to it.

Line 230: The change in FCO2 is indeed small, but on long timescales it could still affect the ocean carbon sink capacity quite a bit (after all, if SST had marginal impact on CO2 flux then there wouldn't be worries about the solubility pump and therefore the ocean carbon sink declining with global warming). However, in equilibrium runs with relatively stable SATs/SSTs as in this study this isn't so important, and the maximum SST difference from your main runs yields a far smaller change. Still, it might be beneficial to calculate how much a 0.21% increase in FCO2 would affect carbon reservoir size over the duration of your runs, if only to demonstrate it's only a minor factor.

In this study, the changes in heat budget are small therefore the effect of SST on FCO2 is not important. Between our simulations, the maximum change of FCO2 is 0.21%. The table below indicates that a 0.21% increase in FCO2 slightly affects the carbon reservoirs in the simulation Bio.

| Carbon reservoir | Effect |
|---|---|
| Atm. $CO_2$ concentration | +0.55% |
| DOC | -0.71% |
| POC | -0.04% |
| DIC | -0.32% |
| Carbon biomass | -0.01% |
| $CaCO_3$ | -0.001% |

Line 234-235: But what's the implication of the larger impact of changed wind speed baseline on FCO2 for your results then? At the moment you use the comparison to imply SST's effect is negligible, but don't go in to the implications of your wind speed sensitivity calculation.

Between all the simulations, the wind field is prescribed and identical. As a consequence, the effect of wind on FCO2 is identical for all the simulations. We just modify the wind speed for the sensitivity analysis. In the "model setup" section, we add sentences to explain the wind fields.

Line 237-238: Additionally cGEnIE/ecoGEnIE is rather low resolution anyway and biota can't move between grid cells, so will not resolve some key regional processes even if seasonality was included.

We agree and add this argumentation in the manuscript.

Line 246: The CARB simulation is not described in Section 5.1, presumably because it's not so interesting for examining ocean dynamics (as you mention in outlining the scenarios), but this could bear repeating here even if briefly (much like CARB is only briefly mentioned in 5.2.1 to state atmospheric CO2 is the same as BioLA).

We agree and will add a brief description.

Line 249-251: I don't think this particular mechanism (PLA increasing chlorophyll biomass via changes in OM export) has been described in this manuscript prior to this point, and as readers haven't necessarily read the cited papers it would be therefore useful to explain this process here in a little more detail. Is it because activating PLA brings production closer to the surface and therefore boosts surface nutrient recycling and surface biomass? Also, "shallower downward flux of OM" is slightly confusing phrasing.

We agree and will rephrase this part.

Line 253-255: It would be useful to provide those previous estimates here for the reader to compare the current estimate with. Looking at Asselot et al. (2021), is the comparable number there 0.45C? If so it's quite a big difference, and emphasises the point earlier that no-seasonality may not affect the overall pattern of findings but will still affect your results to some degree (which you've usefully stated here).

We add the values of previous modelling studies. Due to PLA, we find a global increase of SST of 0.08°C while previous studies find a global SST increase between 0.45-1°C. We mention in the manuscript that the overall pattern of findings is not affected but quantitative changes arise. Since we are interested in a qualitative assessment only, we still find that the non-seasonality is an adequate simplification in the model setup.

Line 259-260: The impacts of stronger overturning fit with your results, but why is overturning a whole Sverdrup stronger in HEAT if the CO2->SAT forcing is the same? Needs some explanation, as a ~13% increase is quite a big difference.

In the simulation HEAT the SST is lower than in the simulation Bio and also the sea-ice cover is slightly higher (9.91% versus 9.79% in Bio), so there is more deep water formation in high latitudes inducing a stronger AMOC. We will add this in the manuscript.

Line 268-281: The differences between HCorg and HCorgSol solubility and PO4/chl/SST are explained in this paragraph, but not why the HCorgX scenarios in general have higher chl biomass & SST than the reference run. Presumably as with HCorg vs. HCorgSol it might be explained by higher PO4 in HCorg vs. Bio (which leads to higher biomass and therefore higher SST due to PLA), but an explanation of how these differences versus the default simulation arise (and indeed why PO4 is higher, explaining in more detail the role of the biological pump here) would be useful at the outset of this paragraph to guide the reader along. I'm also wondering to what extent using only one size class each for phyto- and zooplankton might affect these HCorgX simulations, given the role different size classes play in POM vs. DOM production.

The surface chlorophyll biomass in the simulations HCorgX is higher than the surface chlorophyll biomass in the reference simulation Bio due to the higher surface nutrient concentrations; in Bio the surface PO4 concentration is $0.18 \times 10^{-6}$ mol/kg while in the simulations HCorgX the surface phosphate concentration is $>0.21 \times 10^{-6}$ mol/kg. The higher surface PO4 concentrations in HCorgX in Bio are mainly due to shallower remineralization length scale leading to less export production into the deeper layer (see also Asselot et al. 2021). Higher nutrient concentrations in turn increase the surface chlorophyll biomass enhancing the effect of phytoplankton light absorption. These explanations are added to the revised manuscript.
Please note that we addressed the question concerning the effects of ecosystem complexity, here defined as the number of size classes of phyto- and zooplankton already in the published article of Asselot et al. (2021). We therefore refrain from doing similar experiments since we do not expect fundamentally different results.

Line 289: Would be useful to state what the previous estimate of BioLA vs. Bio CO2 was to avoid the reader having to flick back to Asselot et al. (2021) to compare it themselves.

We will add the value of Asselot et al. (2021).

Line 295-297: What do you mean by more "important" here – higher, I guess? A bit unclear phrasing to me as it stands (similarly in final paragraph sentence.)

Changed.

Line 298: The remineralisation rate would only be higher if temperature-dependent remineralisation (TDR) is activated, otherwise remin. rates follow a fixed profile in cGEnIE & ecoGEnIE (see Crichton et al., 2021 & Armstrong McKay et al., 2021 for discussion of TDR in cGEniE/ecoGEnIE). If the former is the case in this study then you need to state this in the Methods, but if not then higher remin. rates can't be the cause of increased dissolved CO2 – instead it'd most likely be higher production rates leading to more remin. despite remin. being at the same rate. I suspect this might be what you meant here anyway – if so, should be clear on terminology here to avoid confusion.

We agree and will change the sentence accordingly; a temperature-dependent remineralisation is not considered in our model setup.

Line 397-398: You should state the mechanism for how PLA increases atmospheric CO2 here (i.e. PLA = surface warming = lower solubility = more atm. CO2) for clarity.

A sentence will be added to explain how PLA increases the atmospheric CO2 concentration.

Line 332-333: Highest as in least negative, but arguably a tad confusing phrasing as Bio/CARB as plotted have the smallest (negative) bars. Might be better to rephrase sentence starting in line 331 as "A more negative value of net longwave heat flux indicates a greater loss of heat to outer space" along with next sentence accordingly.

We thank for the advice and will rephrase this sentence.

Line 334: Refer the reader here to the next section for details on specific humidity (otherwise it feels like a skipped detail).

We agree and will refer the reader to the adequate figure.

Line 339: This might be my more limited experience of climate thermodynamics relative to biogeochemistry here, but presumably in Bio the same amount of light enters the ocean but is absorbed less close to the surface (as no PLA), so activating PLA is not so much an "additional heat source" as heat absorbed closer to the surface in a way that affects the atmosphere on shorter timescales? If so then on longer ocean overturning and equilibration timescales could total ocean heat uptake in Bio eventually lead to higher outgoing radiation than in this spin-up?

There are no biologically-induced albedo changes and the outgoing radiation is similar – if this is what the reviewer is referring to. Here, we wanted to point out that PLA is an "additional heat source" for the surface of the ocean, where the air-sea heat exchanges occur. We will revise our sentence.

Line 358: How much lower than previous values? Useful to state these things directly so the reader can easily see the difference.

We estimate that phytoplankton light absorption raises the specific humidity by 0.5% while previous estimates indicate an increase of 2-6% (Patara et al., 2012). We will add the previous estimates in the revised manuscript.

Line 362: Confusingly phrased (had to re-read a few times to get it) – might be clearer rephrased as "with lower humidity leading to higher evaporation rates" or similar.

Changed.

Line 374: As mentioned earlier, it's useful to specify what these previous estimates are for easy inter-comparison by the reader. In this case, if the relevant number from Asselot et al. (2021) is 0.45C then arguably in the next sentence ~25% of that value is not such a small difference.

The previous estimates are added to the manuscript.

Line 380-381: Presumably the global net longwave heat flux decreases because of the lower SST.

Indeed, the SST is lower in HEAT compared to Bio, leading to a decrease of the global net longwave flux.

Line 412-417: I'd add that bringing in more plankton size classes (as well as more PFTs) would be useful for more complex ecological dynamics to emerge, which as well as enabling TDR could significantly affect the biological pump pathway. Additionally assessing if there's climate state-dependence, reflecting my earlier comments about the relatively low CO2 baseline. These could of course make nice sensitivity analyses for this present study, but if that's not possible in the time available then they'd certainly be useful next steps (and arguably increasing ecological complexity is a priority ahead of increasing atmospheric complexity with PLASIM). I'd also be interested how this would affect a transient warming simulation – would it likely amplify or dampen global warming? This could be something to draw out as wider implications too.

As suggested by the reviewer, we will add further perspectives on these issues in the final paragraph of the manuscript. Yet, previous studies have already analysed the effect of PLA under a transient warming scenario (Park et al., 2015; Paulsen, 2018). Paulsen (2018) indicates that PLA amplifies SST increase by 0.7K. Additionally, Park et al. (2015) evidence that PLA amplifies future Arctic warming by 20%.

Figure 3: The simulation names might do with being bigger (and/or in top-left corners), as I didn't notice them at first. Also, currently the dotted lines initially seemed to imply that the prescribed pathways are the same in each case, whereas for example the pCO2->SAT prescription in HEAT is from Bio and in CARB is from BioLA. Maybe could add a label above prescribed lines indicating from which setup the prescription is coming from? It'd make the figure busier, but would also make it more self-standing without needing to cross-reference to the text so much.

We will change Figure 3 and the names are clearer in the revised manuscript. However, we would like to keep the figures simple but we will add the information in the figure legends.

Thanks again for your comprehensive review and very valuable comments!

Dr. David A. McKay

27/8/2021

References:

K. A. Crichton, J. D. Wilson, A. Ridgwell, P. N. Pearson, Calibration of temperature-dependent ocean microbial processes in the cGENIE.muffin (v0.9.13) Earth system model. Geosci. Model Dev. 14, 125–149 (2021).

D. I. Armstrong McKay, S. E. Cornell, K. Richardson, J. Rockström, Resolving ecological feedbacks on the ocean carbon sink in Earth system models. Earth Syst. Dyn. 12, 797–818 (2021).

Guihou et al. "Enhanced Atlantic meridional overturning circulation supports the last glacial inception." Quaternary Science Reviews 30.13-14 (2011): 1576-1582.

Park et al. "Amplified Arctic warming by phytoplankton under greenhouse warming." Proceedings of the National Academy of Sciences 112.19 (2015): 5921-5926.

Patara et al. "Global response to solar radiation absorbed by phytoplankton in a coupled climate model." Climate dynamics 39.7 (2012): 1951-1968.

Paulsen. The effects of marine nitrogen-fixing cyanobacteria on ocean biogeochemistry and climate–an Earth system model perspective. Diss. Universität Hamburg, Hamburg, 2018.

---

## Author Comment (AC3)

Alexandre Pohl (Biogéosciences, Dijon, FRANCE)
02 September 2021

**Summary:**

Asselot et al. study the mechanisms by which light absorption by the phytoplankton impacts the ocean atmosphere system, global temperatures in particular. They use the Earth System Model of intermediate complexity GENIE, extended to include light absorption by Asselot et al. (2021), and conduct several numerical experiments by turning on and off key feedbacks. This approach allows the authors to quantify the impact of the different mechanisms studied. Asselot et al. conclude that the air-sea CO2 exchange has a much larger impact on biologically-induced global climate warming than changes in the heat fluxes.

We would like to thank Alexandre Pohl for his constructive comments.

**General evaluation:**

I think that the authors approach an interesting topic. They present a clear modeling strategy. Most of the main text is clear and concise. Figures and Tables are mostly well adapted and satisfactorily convey the authors' message. However, several points should me made clearer. More importantly, I'm worried about key choices made by the authors regarding their modeling setup, which I think are not really obvious and have the potential to significantly impact the results and conclusions.

We revise the manuscript to add clarity. We tried to answer the concerns of the reviewer and to justify our model setup.

**Main comments:**

- Modeling setup:
    - Model spinup and equilibrium: Based on Section 3, the authors ran cGENIE with biogem for 10 kyrs and then restarted the model with ecogem for another 1000 years. First, I don't understand how ecogem can be run based on a biogem-only restart, but this is a technical point. More importantly, I don't think that this setup can lead to robust results. Indeed, changing from biogem to ecogem is expected to lead to important changes. An example is the pCO2 that drops from 278 ppm in the biogem run to 169 ppm in the Bio simulation. Such drastic change is expected to impact global climate and I don't think that 1000 years are enough to reach a new equilibrium. I would instead suggest running ecogem simulations for 10–20 kyrs and make sure that equilibrium is reached. Ensuring a robust equilibrium is particularly important considering the subtle changes reported between the different experiments (e.g., Table 7).

BIOGEM represents the transformation and spatial distribution of elemental biogeochemical tracers, such as nutrient, DOC and POC. This model component needs ~8,500 years until steady state. We run the 10,000-years spinup only with BIOGEM to start the simulations with a realistic nutrient

distribution. ECOGEM considers only the living component that needs only 800 years to reach a steady state. The changes in atmospheric CO2 concentration after 1,000 years are < 0.03 ppm/year, thus we consider the climate system in equilibrium.

o Climate state- and model configuration- dependence of the results: I also don't understand the choice of a cold climate (169 to 178 ppm; Table 5). I do expect the results of the study to be climate state-dependent. It would be interesting to determine if the same conclusions can be reached when using a higher baseline CO2 level (e.g., 350 ppm or above) (which might lead to very different results, due for instance to the lower seaice cover). At least, the authors should state that their results are expected to be climate state-dependent. I also think that it should be made clear that the present-day continental configuration is used. I also expect results to potentially vary with the land-sea mask.

We agree that we should discuss the climate state in our final section and will do this in the revised version. Yet, higher CO2-concentrations will not fundamentally change our conclusions, because higher CO2-concentrations imply higher atmospheric temperature and thus higher SST. The higher SST decreases the CO2 solubility, leading to a larger air-sea CO2 flux. Please note that we do not prescribe the atmospheric CO2-concentrations neither prescribe CO2 emissions but allow the system and the CO2-concentrations to evolve freely.
The present-day continental configuration is used.

o Absence of seasonal cycle: Based on Appendix B, it appears that neglecting the seasonal cycle in the version of ecoGENIE modified to include light absorption leads to a global temperature difference bias (0.33 – 0.14 = 0.19 °C) that is larger than most temperature differences computed in Table 7. I think that the absence of seasonal cycle thus constitutes a major limitation to this work and would encourage the authors to repeat the experiments with seasonality.

For technical reasons the seasonality is removed. In this study, for several simulations, we prescribe a SST field. The prescribed SST field comes from the reference run (Bio) and unfortunately the SST outputs are annual means. As a consequence, it does not make sense to prescribe a yearly-averaged SST field while turning on the seasonal cycle.
Enabling seasonality would lead to larger seasonal increase of temperature but it would also lead to larger seasonal decrease in CO2 solubility. We don't think that the heat-pathway would overrule the CO2-pathway. Enabling seasonality would not change the qualitative importance of each climate pathway. Therefore, we consider that removing the seasonal cycle is an adequate simplification for our study. We would also like to refer to our comments to the 2$^{nd}$ reviewer on this issue.

o Absence of size classes: I don't understand why the authors modified the model of Ward et al. In Appendix A, it is stated that it permits reducing computational time. However, a 10-kyr ecogem model run using the 36x36 grid with 16 vertical levels requires less than 5 days on a single core. Although I understand that using 32 levels probably makes the model more expensive, I guess that the model remains very fast to run and tractable. In any way, please make sure to describe the model in a

consistent manner. For now, it is not very clear: in section 2.1.3, the model of Ward et al. is described as including 2 PFTs with size classes, while in Appendix A, it is described as including 16 PFTs. There is a confusion between the number of plankton types (zoo vs. phyto) and the number of size classes.

We agree and apologize; we will modify the section 2.1.3 and Appendix A for clarification about how many size classes we use. Please note that in a previous paper (Asselot et al., 2021, JAMES), we already analyse the effect of phytoplankton light absorption between simulations with 1 phyto- and 1 zooplankton size classes and simulations with 6 phyto- and 6 zooplankton size classes. Between these two different ecosystem complexities, the effect of phytoplankton light absorption on the climate system is similar. Thus we decided to simplify the ecosystem and only keep 1 phyto- and 1 zooplankton size-classes. Again, we would like to refer to our answer to the 2$^{nd}$ reviewer.

o Temperature-dependent remineralization: Is any temperature-dependent remineralization scheme employed in this study? It is not stated anywhere, but lines 249–251 suggest that the higher SSTs lead to a shallower remineralization. If so, please clarify this point and provide a reference.

The shallower remineralization is due to the higher production with phytoplankton light absorption and due to a larger source to the DOM with a shallower remineralization length scale. We will add the explanation in the revised manuscript.

o Absence of light limitation by sea ice in ecogem: Although it probably has a minor impact, I also note that the attenuation of the photosynthetically available radiation by sea-ice in ecogem, as now part of the muffingen release, does not seem to be used in these experiments.

We thank the reviewer for this comment. Indeed, the light limitation by sea-ice in ECOGEM is not included in our simulations. This clarification will be added to the revised manuscript.

o Absence of dynamical atmosphere: Based on Section 4.2, it seems that changing wind stress could have a major impact on the results. cGENIE wind fields are boundary conditions and do not vary with changing climate. I agree with the authors that a dynamical atmosphere would be useful (lines 412–413) but would rather present this as a current limitation / necessary next step rather than a possible way forward and encourage the authors to expand on this point.

Yes, in our simulations, the wind fields are prescribed thus the wind field is similar between simulations. We agree that we should make clear that our discussion is rather an outlook. Previous studies evidence either an increase in wind speed in the subpolar regions (Patara et al., 2012, Climate Dynamics) or an enhanced atmospheric dynamics (Gnanadesikan et al., 2009, Journal of Phy. Oceano.; Wetzel et al., 2006, Journal of Climate) due to phytoplankton light absorption. Therefore, with a dynamical atmospheric component, the wind speed would increase due to phytoplankton light absorption. As a result, the air-sea $CO_2$ flux increases as well. One indeed could speculate that

including a dynamical atmosphere would reinforce the importance of the CO2-pathway. We will make it more clear and will add these aspects in the revised manuscript.

• Model description: In section 2.1.2, the authors should clearly state that the description only refers to their model setup / the choices that they made. For instance, all productivity schemes of cGENIE do not include light nor iron limitation.

We will better explain our specific model setup.

• "Previous estimates": Please provide the correspond values, on lines 252, 253, 289, 374.

We add the previous estimates for a better comparison with previous studies.

**Other (mostly minor) points:**

• Line 8: "dissolved CO2" > "air-sea CO2 flux"?
Changed.

• Line 59: "due to fluctuations"
Changed.

• Line 62: "as follows"
Changed.

• Line 71: "composed"
Changed.

• Line 77: "cGENIE"
Changed.

• Line 120: state variables for iron, too?
Changed.

• Line 139: "(Eq. 2)"
Changed.

• Lines 142–144: I don't follow this. It is stated that "the whole light absorption leads to heating of the water". Shouldn't the fraction used by the plankton be subtracted, according to line 142?
We add clarification by stating: "Part of the light absorbed is used by phytoplankton for photosynthesis and part leads to heating of the water."

• Line 148: "(Eq. 3)"
Changed.

• Line 152: delete "certain"?
Changed.

• Line 162: "(Eq. 4)"
Changed.

• Lines 200–201: I thought that atmospheric pCO2 was prescribed, based on lines 199–200?
No, for this simulation CARB, we force the atmosphere with the heat fluxes from the reference simulation Bio and with the atmospheric CO2 concentration from the simulation BioLA. Between CARB and Bio the heat fluxes are similar but the atmospheric CO2 concentrations differ. Thus the simulation CARB permits to understand the effect of phytoplankton-induced changes in atmospheric CO2 concentration.

• Section 4.1: I don't understand the utility of this section (which should be called "climate sensitivity" by the way?). Delete?
We put the whole section 4 in the appendix.

• Line 216: "differs"
Changed.

• Line 227: "analyses"
Changed.

• Line 228: "do not exceed the maximum difference of SST between our simulation results". I don't understand this.
We remove this part of the sentence.

• Line 230: "Even larger SST fluctuations … interface". Not shown?
This sentence refers to the previous sentence where we increase the SST by 1°C while the air-sea CO2 flux only increases by 2.58%. We will move this part in Appendix.

• Lines 231–232: "We increase the wind speed by 0.2 m/s… (Knutson and Tuleya, 2004)." Please expand. It would be useful to the reader.
We will add more explanation.

• Line 270: I would avoid calling chlorophyll concentration a "climate variable".
This study and several previous studies focusing on biogeochemical and biogeophysical mechanisms evidence that phytoplankton (and thus chlorophyll biomass) influences the climate system. Thus we consider chlorophyll being a "climate variable".

• Line 277: Is this difference larger than the one obtained when running the same experiment twice?
Running the simulations twice does not change our results.

• Line 296: "biomass is more important in HEAT than in the reference simulation (Table 4)"
Changed.

• Lines 308–310: please rephrase or add punctuation.
Changed.

• Line 331: "a higher negative value" > "lower absolute value" may be clearer?
We change the sentence by: "a more negative value…".

• Lines 337, 368: please delete "rather".
Changed.

• Lines 351: "in these simulations"?
Changed.

• Line 357: "Oschlies (2004) and Lengaigne et al. (2009)"?
Changed.

• Lines 364–365: "specific humidity and evaporation are higher in the simulation CARB than in BioLA"
Changed.

• Line 371: please delete "indubitably".
Changed.

• Line 401: formatting of the references.
Changed.

• Lines 413–415: "Observations… […] Wurl et al. (2018)." Please delete or expand. As such, the message is not easy to understand.
Changed.

• Appendix B1: should refer to Table B1.
Changed.

• Line 435: "decreases the mean annual SST"
Changed.

• Line 437: "dampens"
Changed.

• Lines 444–445: "maximum global sea-ice cover change (or difference)"
Changed.

• Caption of Fig. 2: EMBM should be defined.
The names of the different model components are now defined in the main text.

• Figure 5: Unless I missed something, this figure can be deleted since it is redundant with Fig 3 and the 3rd column of Table 7.
We will change this figure and will remove Table 7.

• Table 1: "Bio – Reference run without phytoplankton light absorption".
Changed.

---

## Referee Report (RR1)

**Review 2 of "*Climate pathways behind phytoplankton-induced atmospheric warming*" by Asselot et al. (2021) for Biogeosciences**

The authors have carefully revised their manuscript in response to the reviewer comments, and I believe the manuscript has improved substantially as a result and is ready to be published with a few minor revisions. Below I focus on the responses to my own comments, but as they overlapped substantially with the other reviewers I believe their queries have also been largely covered.

The Abstract has been clarified as suggested, with more detail provided on what has been done and why along with key results. The Introduction also now has more background on how PLA affects the key pathways analysed in this study. My only suggestion is an additional statement to clarify how you go about disentangling the mechanisms in this study (see below for detail).

More detail has been provided throughout the rest of manuscript as well, for example how these results directly relate to previous numbers, mechanism explanations from Asselot et al. (2021), and methodological details for e.g. the new light scheme as well as more model description, with just a few more useful clarifications suggested below.

The sensitivity analysis of repeating Bio & BioLA with multiple size classes was performed as suggested, showing a small but non-negligible effect as suspected. The authors have included this in the Appendix which I think is worthwhile, although I'd suggest referring to this Appendix more clearly in the main text (see below). Reducing the main text dedicated to the sensitivity analyses is also wise, as is introducing a specific Limitations section.

I still have some reservations about the low CO2 baseline and how the results may quantitatively change with climate state, but the authors have now made this limitation and that they're focusing on the relative, qualitative changes clearer throughout the manuscript (although I suggest a little extra detail on this below).

Other questions I had regarding the AMOC strength in the HEAT simulation, drivers of HCorg vs. Bio biomass differences, temperature-dependence, and size classes have also been answered and clarified in the text, along with minor issues rectified and some more implications & further work suggestions in the Conclusion.

**Specific comments (line numbers refer to track changes document):**

Line 69: I think this final Introduction paragraph could do with an extra sentence or two at the end to state clearly how you achieve the disentangling mentioned in the first sentence, i.e. that in this study you effectively turn on and off specific pathways by prescribing values in order to isolate the impact of the 3 different mechanisms you've just described above. Additionally, the Introduction's end here feels a bit sudden now that it ends on the 3 mechanisms and the original final paragraph with paper structure has been deleted, so the above suggestion might help to round off the Introduction.

Lines 86-89: I did indeed ask for a little more explanation of this citation here, but this is perhaps more than necessary and somewhat disrupts the text flow in this paragraph. Feel free to reduce this to a briefer, within-sentence summary if you prefer – for example: "*Additionally, cGEnIE **has been employed to assess the sensitivity of atmospheric CO2 to biogeochemical pumps, ocean circulation and climate feedbacks **in the Southern Ocean** (Cameron et al., 2005)**. A new ecosystem component…*"

Line 126: Might be worth clarifying that labile organic matter is mostly but not entirely remineralised in the upper water column, as it follows a fixed profile (specifically the Martin Curve) that declines geometrically until reaching a small (but not zero) asymptotic flux by 1000m.

Lines 127-128: Slightly confusing phrasing here – could potentially be misread as CaCO3 turns into remineralised POM. Presumably actually means PIC remineralisation follows the same profile as POC

(i.e. Martin Curve): could rephrase as *"…and its dissolution below the surface **follows the same profile** as the remineralisation of POM"*. It might also be worth mentioning that CaCO3 production is parameterised as a fixed ratio to POC production, and so is not independent in this model (at least not until a separate calcifier PFT is implemented).

Line 210: This would be a good place to briefly mention your model setup has a reasonable match to observations (as you state well in your response doc on pg14), before referring the reader to Asselot et al. (2021) for further details (you mention earlier that cGEnIE has been calibrated to observations, but some readers will want reassurance that it's still good with this new light parametrisation).

Line 211: It would be useful to mention here that you've done a related sensitivity analysis and directly link to table B1, e.g. something like: "*For simplification, only one phytoplankton and one zooplankton species are included in the model setup (Appendix A1 and B1). **Repeating our main simulations with multiple size classes for each results in relatively small differences (Table B1).**"*

Line 212: I'd suggest a little more detail in this sentence to clarify that unlike your previous study prescribing SST (and therefore removing seasonality) is necessary for your experimental design in this study – your response has helped me to understand this bit now, but given that Reviewer 3 also got a bit confused on this it's likely some readers will also need a little extra guidance on this.

Line 220: The term "*climate sensitivity*" here might get mixed up with the concept of Equilibrium Climate Sensitivity (ECS) by readers, which in this model is effectively fixed – I think what you mean is sensitivity of your variables of interest (absolutes of SAT, SST, chlorophyll) to CO2, rather than the relative change in SAT per doubling of CO2 (ECS).

Line: 291: Upward vertical velocity of what, exactly? (also line 322)

Lines 343-345: It's good that you explicitly discuss the model's low CO2 baseline and its causes here, but I think you should also include some of your discussion from the response doc re. how this might affect your results (i.e. pg14 starting where you say "*We agree that our quantitative estimates would be affected if the atmospheric CO2 concentration of the reference run would be higher but…*"), either in this section or in the new Limitations section.

Lines 471-490: This paragraph could do with breaking up in to two or more for clarity.

Lines 477-480: Could do with citation(s) here re. PLA, denitrification, and hypoxia if available (if not, indicate it's a suggestion). Could also do with making the hypoxia -> denitrification step explicit.

Line 480: Is this increase just from the PLA, similar to this study, or because of adding land as well?

Line 532: I know this is not what you mean here, but this could be misread as introducing size classes has no impact on the climate when in fact here it is just less impact in this case than PLA.

Lines 547-548: Could add something like: "***, indicating that surface chlorophyll biomass is more "sensitive" to PLA than an increase of 40 ppm in pCO2***" from your response doc at the end of this sentence in order to make implication clear.

Lines 563-564: Maybe add what sort of order the slight effect is for clarity, e.g. "*This small increase slightly affects the carbon reservoirs in our simulations **by < 1%**"*.

*Dr. David A. M$^c$Kay, 19/11/2021*

---

## Author Response (AR3)

**Review 2 of "Climate pathways behind phytoplankton-induced atmospheric warming" by Asselot et al. (2021) for Biogeosciences**

The authors have carefully revised their manuscript in response to the reviewer comments, and I believe the manuscript has improved substantially as a result and is ready to be published with a few minor revisions. Below I focus on the responses to my own comments, but as they overlapped substantially with the other reviewers I believe their queries have also been largely covered.

We thank Dr. David A. McKay for the time he invested to review our manuscript and for his suggestions. We would also thank the associate editor for his interest in our study. We address all the suggestions of the reviewer in the "specific comments" section.

The Abstract has been clarified as suggested, with more detail provided on what has been done and why along with key results. The Introduction also now has more background on how PLA affects the key pathways analysed in this study. My only suggestion is an additional statement to clarify how you go about disentangling the mechanisms in this study (see below for detail).

More detail has been provided throughout the rest of manuscript as well, for example how these results directly relate to previous numbers, mechanism explanations from Asselot et al. (2021), and methodological details for e.g. the new light scheme as well as more model description, with just a few more useful clarifications suggested below.

The sensitivity analysis of repeating Bio & BioLA with multiple size classes was performed as suggested, showing a small but non-negligible effect as suspected. The authors have included this in the Appendix which I think is worthwhile, although I'd suggest referring to this Appendix more clearly in the main text (see below). Reducing the main text dedicated to the sensitivity analyses is also wise, as is introducing a specific Limitations section.

I still have some reservations about the low CO2 baseline and how the results may quantitatively change with climate state, but the authors have now made this limitation and that they're focusing on the relative, qualitative changes clearer throughout the manuscript (although I suggest a little extra detail on this below).

Other questions I had regarding the AMOC strength in the HEAT simulation, drivers of HCorg vs. Bio biomass differences, temperature-dependence, and size classes have also been answered and clarified in the text, along with minor issues rectified and some more implications & further work suggestions in the Conclusion.

**Specific comments (line numbers refer to track changes document):**

Line 69: I think this final Introduction paragraph could do with an extra sentence or two at the end to state clearly how you achieve the disentangling mentioned in the first sentence, i.e. that in this study you effectively turn on and off specific pathways by prescribing values in order to isolate the impact of the 3 different mechanisms you've just described above. Additionally, the Introduction's end here

feels a bit sudden now that it ends on the 3 mechanisms and the original final paragraph with paper structure has been deleted, so the above suggestion might help to round off the Introduction.

We added a sentence at the end of the introduction section. The sentence is: "To achieve the disentangling of the specific climate pathways, we turn on and off the climate pathways by prescribing values in our ESM in order to isolate their impact on the climate system."

Lines 86-89: I did indeed ask for a little more explanation of this citation here, but this is perhaps more than necessary and somewhat disrupts the text flow in this paragraph. Feel free to reduce this to a briefer, within-sentence summary if you prefer – for example: "Additionally, cGEnIE **has been** employed to assess the sensitivity of atmospheric CO2 to biogeochemical pumps, ocean circulation and climate feedbacks in the **Southern Ocean** (Cameron et al., 2005). A new ecosystem component…"

We remove the sentence as suggested by the reviewer.

Line 126: Might be worth clarifying that labile organic matter is mostly but not entirely remineralised in the upper water column, as it follows a fixed profile (specifically the Martin Curve) that declines geometrically until reaching a small (but not zero) asymptotic flux by 1000m.

We clarified this point and added: "and labile organic matter **mostly** remineralised in the upper water column".

Lines 127-128: Slightly confusing phrasing here – could potentially be misread as CaCO3 turns into remineralised POM. Presumably actually means PIC remineralisation **follows the same profile** as POC 2 (i.e. Martin Curve): could rephrase as "…and its dissolution below the surface follows the same profile as the remineralisation of POM". It might also be worth mentioning that CaCO3 production is parameterised as a fixed ratio to POC production, and so is not independent in this model (at least not until a separate calcifier PFT is implemented).

We cease the suggestion of the reviewer and changed the sentence. Furthermore, we also added a sentence explaining that CaCO3 production is parameterised as a fixed ratio to POC production.

Line 210: This would be a good place to briefly mention your model setup has a reasonable match to observations (as you state well in your response doc on pg14), before referring the reader to Asselot et al. (2021) for further details (you mention earlier that cGEnIE has been calibrated to observations, but some readers will want reassurance that it's still good with this new light parametrisation).

We added the sentence "Our model setup has a reasonable match to observations and further details can be found in Asselot et al., 2021".

Line 211: It would be useful to mention here that you've done a related sensitivity analysis and directly link to table B1, e.g. something like: "For simplification, only one phytoplankton and one zooplankton species are included in the model setup (Appendix A1 and B1). **Repeating our main simulations with multiple size classes for each results in relatively small differences (Table B1).**"

We added the sentence "Repeating our main simulations with multiple size classes leads in small differences compared to the simulations with one size class".

Line 212: I'd suggest a little more detail in this sentence to clarify that unlike your previous study prescribing SST (and therefore removing seasonality) is necessary for your experimental design in this study – your response has helped me to understand this bit now, but given that Reviewer 3 also got a bit confused on this it's likely some readers will also need a little extra guidance on this.

We added more details to explain why we remove the seasonal cycle.

Line 220: The term "climate sensitivity" here might get mixed up with the concept of Equilibrium Climate Sensitivity (ECS) by readers, which in this model is effectively fixed – I think what you mean is sensitivity of your variables of interest (absolutes of SAT, SST, chlorophyll) to CO2, rather than the relative change in SAT per doubling of CO2 (ECS).

Indeed, we meant sensitivity of our climate variables. We changed this sentence.

Line: 291: Upward vertical velocity of what, exactly? (also line 322)

We meant that the upward vertical velocity in the upwelling regions is enhanced. We revised the sentence.

Lines 343-345: It's good that you explicitly discuss the model's low CO2 baseline and its causes here, but I think you should also include some of your discussion from the response doc re. how this might affect your results (i.e. pg14 starting where you say "*We agree that our quantitative estimates would be affected if the atmospheric CO2 concentration of the reference run would be higher but…*"), either in this section or in the new Limitations section.

We added a discussion of the low CO2 baseline in the "limitations" section.

Lines 471-490: This paragraph could do with breaking up in to two or more for clarity.

This paragraph has been broken up in two but the formatting will be made by the publisher.

Lines 477-480: Could do with citation(s) here re. PLA, denitrification, and hypoxia if available (if not, indicate it's a suggestion). Could also do with making the hypoxia -> denitrification step explicit.

After literature research, no studies have investigated the link between PLA, denitrification and hypoxia. We changed the sentences and added suggestion on this part. We also added a link between hypoxia and denitrification.

Line 480: Is this increase just from the PLA, similar to this study, or because of adding land as well?

The increase in heat budget is due to PLA. We revised our sentence.

Line 532: I know this is not what you mean here, but this could be misread as introducing size classes has no impact on the climate when in fact here it is just less impact in this case than PLA.

We revised our sentence: "We show that introducing more size classes has a smaller effect on the climate system than phytoplankton light absorption"

Lines 547-548: Could add something like: **", indicating that surface chlorophyll biomass is more "sensitive" to PLA than an increase of 40 ppm in pCO2**" from your response doc at the end of this sentence in order to make implication clear.

As suggested by the reviewer, we revised our sentence.

Lines 563-564: Maybe add what sort of order the slight effect is for clarity, e.g. "This small increase slightly affects the carbon reservoirs in our simulations **by < 1%**".

We revised our sentence as suggested by the reviewer.

Dr. David A. McKay, 19/11/2021